# Leveraging Spatial and Temporal Correlations in Sparsified Mean Estimation

**Divyansh Jhunjhunwala**
Carnegie Mellon University
Pittsburgh, PA 15213
djhunjhu@andrew.cmu.edu

**Ankur Mallick**
Carnegie Mellon University
Pittsburgh, PA 15213
amallic1@andrew.cmu.edu

**Advait Gadhikar**
Carnegie Mellon University
Pittsburgh, PA 15213
agadhika@andrew.cmu.edu

**Swanand Kadhe**
University of California Berkeley
Berkeley, CA 94720
swanand.kadhe@berkeley.edu

**Gauri Joshi**
Carnegie Mellon University
Pittsburgh, PA 15213
gaurij@andrew.cmu.edu

## Abstract

We study the problem of estimating at a central server the mean of a set of vectors distributed across several nodes (one vector per node). When the vectors are high-dimensional, the communication cost of sending entire vectors may be prohibitive, and it may be imperative for them to use sparsification techniques. While most existing work on sparsified mean estimation is agnostic to the characteristics of the data vectors, in many practical applications such as federated learning, there may be spatial correlations (similarities in the vectors sent by different nodes) or temporal correlations (similarities in the data sent by a single node over different iterations of the algorithm) in the data vectors. We leverage these correlations by simply modifying the decoding method used by the server to estimate the mean. We provide an analysis of the resulting estimation error as well as experiments for PCA, K-Means and Logistic Regression, which show that our estimators consistently outperform more sophisticated and expensive sparsification methods.

## 1   Introduction

Estimating the mean of a set of vectors collected from a distributed system of nodes is a subroutine that is at the core of many distributed data science applications. For instance, a fusion server performing inference may seek to estimate the mean of environmental measurements made by geographically distributed sensor nodes. While inference may be a one-shot task, model training is often inherently iterative in nature. For instance, model training using a federated learning framework [19, 35] is divided into communication rounds, where in each round, the central server has to estimate the mean of model updates or gradients sent by edge clients. Other examples of such iterative methods include principal component analysis (PCA) using the power iteration methods and K-Means.

In most such distributed learning and optimization applications, the nodes sending the data are highly resource-constrained and bandwidth-limited edge devices. Hence, sending high-dimensional vectors can be prohibitively expensive and slow. A simple yet effective solution to reduce the communication

35th Conference on Neural Information Processing Systems (NeurIPS 2021).

cost is to send a sparsified vector containing only a subset of the elements of the original vector [30]. The Rand-$k$ sparsification scheme chooses a subset of $k$ elements uniformly at random, while the Top-$k$ scheme selects the $k$ highest magnitude indices. Other more advanced schemes such as [34, 36] optimize the weights assigned by a node to different elements of its vector when sparsifying it and sending it to the central server.

A common thread in the existing techniques [36, 34, 21] is that they are agnostic to the fact that in many practical applications, the vectors sent by the nodes are correlated across different nodes and over consecutive rounds of iterative algorithms that use mean estimation as a subroutine. For example, in a sensor data fusion applications, sensor nodes in close promixity will make similar environmental measurements and their resulting data vectors will be correlated. In a federated learning application, model updates sent by a node tend to be similar across consecutive communication rounds. The focus of this work is to take into account these spatial and temporal correlations when combining sparsified vectors sent by nodes and estimating their mean.

## 1.1  Main Contributions

In this work, we carefully design the aggregation method used by the central server to estimate the mean of sparsified vectors received from different nodes. This approach does not introduce additional memory requirement or computation at the nodes. As a result, our proposed mean estimators are compatible with any sparsification method used by the nodes. For the purpose of this paper and the theoretical analysis of the estimation error, we consider random sparsification at the nodes and focus on unbiased mean estimation. However, other sparsification techniques including Top-$k$ sparsification and the methods proposed in [34, 36] can also be used by the nodes, albeit at the cost of additional local computation. Finally, while our goal is to leverage temporal and spatial correlations, we present methods that do not explicitly need information about the correlation; the estimators only consider the fact that such correlation exists. By following this design philosophy of requiring little or no side information and additional local computation, this paper makes the following contributions:

1. *Leveraging Spatial Correlation:* In random sparsification, a randomly chosen subset of $k$ out of $d$ elements of the vector is sent by a node to the central server. The standard approach is for the server to decode via simple averaging of the vectors assuming the missing values to be zero, and scaling with an appropriate constant to get an unbiased mean estimate. Instead, in Section 3, we propose a family of estimators called Rand-$k$-Spatial that adjust the weights assigned to different vectors according to a parameter that represents the amount of spatial correlation. When the correlation parameter is not known, we propose an *average-case* estimator that works well in practice.

2. *Leveraging Temporal Correlation:* If the vectors sent by a node are similar across iterations, previously sent elements of the vectors can be used to *fill in* the missing elements of a sparsified vector. In Section 4, we build on this idea of leveraging these temporal correlations and propose the Rand-$k$-Temporal estimator. While our estimator requires additional memory at the central server, it does not add or change any computation or storage at the nodes, which are more likely to be resource-constrained.

Along with designing these estimators and theoretically analyzing their mean squared error, in Section 5 we present experiments showing that the estimator error can be drastically reduced when there are spatial and temporal correlations.

## 1.2  Related Work

The problem of communication-efficient mean estimation when the data is distributed across several nodes is considered in [40, 13, 7, 33, 10, 24]. Among them, [40, 13, 7] consider statistical mean estimation assuming i.i.d. data distribution across nodes. Works [33, 10, 24, 17] consider empirical mean estimation without assuming any statistical distribution on the data, and quantize the vectors to a small number of bits. There is a significant recent interest in considering the communication-efficient (empirical) mean estimation problem in the context of distributed stochastic gradient descent, see e.g., [1, 2, 12, 5, 37, 36, 34, 20, 4, 23, 6, 14, 27]. These works can broadly be partitioned into three categories: (i) Quantization: encoding each element of the vectors to a small number of bits [18, 1, 12, 5, 37, 20, 23, 27], (ii) Sparsification: sending only a subset of elements of the vectors [2, 34, 36]. (iii) Generic compression operators: defining a class of compressors (with specific properties such as unbiasedness, bounded variance) that encompass quantization and sparsification as

special cases [20, 14]. We focus our attention to sparsification since it is orthogonal to quantization and the two can also be combined, see e.g., [4]. To the best of our knowledge, spatial correlation across nodes has not yet been considered in the context of quantized and sparsified mean estimation. Temporal correlation has been considered in some recent sparsification works [27, 14] and error-feedback [20, 30]. However, these methods change the sparsification method at the nodes and require additional memory and local computation. In contrast, we only modify the server-side aggregation method. See Section 4 for more details of prior work on exploiting temporal correlation.

## 2  Problem Formulation

**System Model and Notation.** We consider a setup consisting of $n$ geographically distributed nodes, and a central server. Each node generates a $d-$dimensional vector $\mathbf{x}_i = [x_{i1}, \ldots, x_{id}]^T$ for $i = 1, \ldots, n$, where $x_{ij}$ denotes the $j^{\text{th}}$ element of node $i$'s vector $\mathbf{x}_i$. Note that we do not assume any specific distribution from which these vectors are generated. Spatial correlations between the vectors in Section 3 are measured using the sum of their inner products. In Section 4 we measure temporal correlations by considering the $L_2$ distance between the vectors sent by a node in different iterations. The central server seeks to estimate the mean of these vectors:

$$\bar{\mathbf{x}} = \frac{1}{n} \sum_{i=1}^{n} \mathbf{x}_i \tag{1}$$

In order to save communication cost, each node sends a sparsified version $\mathbf{h}_i$ of its data vector $\mathbf{x}_i$ to the central server. The estimated mean $\hat{\mathbf{x}}$ is a function of the $\mathbf{h}_1, \ldots, \mathbf{h}_n$ sent by the $n$ nodes.

**Rand-$k$ Sparsified Mean Estimation.** Random sparsification [30] is a simple and commonly used way to sparsify a vector in order to save communication cost. In the Rand-$k$ scheme, node $i$ selects $k$ elements of $\mathbf{x}_i$ uniformly at random and sends these (along with their position information) to the central node. Thus, the $j^{\text{th}}$ element of the sparsified vector $\mathbf{h}_i$ is $h_{ij} = x_{ij}$ if node $i$ sends coordinate $j$ (with probability $k/d$) and is 0 otherwise. The central node computes the mean estimate as

$$\hat{\mathbf{x}} = \frac{1}{n} \frac{d}{k} \sum_{i=1}^{n} \mathbf{h}_i. \tag{2}$$

The scaling factor $\frac{d}{k}$ ensures that $\frac{d}{k} \mathbb{E}[\mathbf{h}_i] = \mathbf{x}_i$ and therefore $\mathbb{E}[\hat{\mathbf{x}}] = \bar{\mathbf{x}}$, i.e., the estimate is unbiased. For the purpose of this paper, we will focus on unbiased random sparsification at the nodes. However, our techniques for leveraging spatial and temporal correlations can be applied to biased sparsification methods such as Top-$k$ as well.

**Estimation Error Metric.** As done in most previous works, we measure the quality of the estimator in terms of the mean squared error (MSE) $\mathbb{E}_{\hat{\mathbf{x}}}\left[\|\hat{\mathbf{x}} - \bar{\mathbf{x}}\|^2\right]$, where the expectation is taken over the choice of the indices that are retained by each of the nodes when sparsifying their data vectors $\mathbf{x}_i$, see, e.g., [33, 12, 10]. The following result, known in folklore, quantifies error in estimating the true mean $\bar{\mathbf{x}}$ (proof in Appendix A).

**Lemma 1** (Rand-$k$ Estimation Error). *The mean squared error (MSE) of estimate $\hat{\mathbf{x}}$ produced by the Rand-$k$ sparsification scheme described above is given by*

$$\mathbb{E}\left[\|\hat{\mathbf{x}} - \bar{\mathbf{x}}\|^2\right] = \frac{1}{n^2}\left(\frac{d}{k} - 1\right) R_1 \tag{3}$$

*where $R_1 = \sum_{i=1}^{n} \|\mathbf{x}_i\|^2$, the sum of the squared magnitudes of the data vectors.*

## 3  Improving Random-$k$ by Leveraging Spatial Correlations

Consider Rand-$k$ sparsification protocol (described in Section 2) where node $i$ sparsifies its vector $\mathbf{x}_i$ to $\mathbf{h}_i$ by retaining only $k$ randomly chosen elements and setting the rest to 0, and sends $\mathbf{h}_i$ to the server. The sampled coordinates are scaled by a factor of $d/k$ to get an unbiased estimate of each node vector. However, consider the case where each node has the same vector, i.e., $\mathbf{x}_1 = \mathbf{x}_2 = \ldots = \mathbf{x}_n$. If $M_j > 0$ nodes send their $j^{\text{th}}$ coordinate, then the $j^{\text{th}}$ coordinate of the mean can be exactly estimated

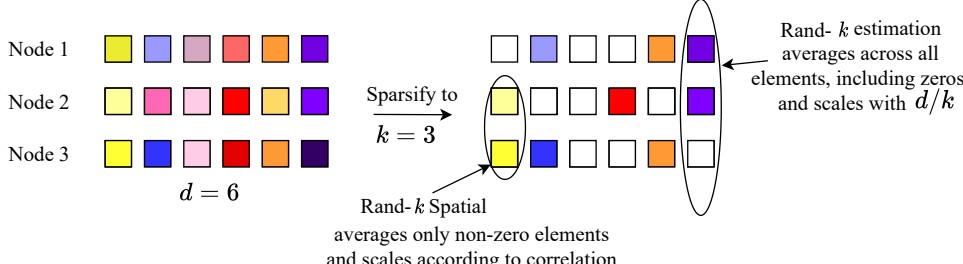

Figure 1: To estimate each element of the mean, the standard Rand-$k$ estimator averages the values received from all nodes, including the zeros. Our rand-$k$-spatial estimator only averages the non-zero elements and scales the average according to the degree of spatial correlation.

as $\hat{x}_j = (\sum_i h_{ij})/M_j$. On the other hand, the scaling factor of $d/k$ in the Rand-$k$ estimator can introduce a large error depending on the value of $M_j$ (e.g., if $M_j = 1$ but $d/k = 10$).

The above example suggests that there are scenarios in which the Rand-$k$ estimator, despite being unbiased, can incur a large MSE. In this section, we will introduce the Rand-$k$-Spatial *family* of unbiased mean estimators (illustrated in Fig. 1) whose members compute mean estimates depending on the amount of correlation between the data at the nodes. We will show that the conventional Rand-$k$ estimator is a member of this family and gives the lowest MSE (among the family) only when the data is uncorrelated, while other members of the family can yield lower values of MSE depending upon the level of correlation. We will also describe how to obtain the estimator from this family that minimizes the MSE when the correlation between the data is known and present an estimator that gives lower MSE than Rand-$k$ on average when correlations are unknown.

**The Rand-$k$-Spatial family of estimators.** Let $M_j$ be the number of nodes that send their $j^{\text{th}}$ coordinate. Observe that $M_j$ is a binomial random variable that takes values in the range $\{0, 1, \ldots, n\}$ with $\Pr(M_j = m) = \binom{n}{m} p^m (1-p)^{n-m}$ for all $j$, where $p = k/d$. Instead of scaling each element by the constant $d/k$ as in Rand-$k$ (cf. (2)), we propose the Rand-$k$-Spatial family of estimators wherein the scaling is computed *as a function of* $M_j$ and the level of spatial correlation between the vectors. In particular, we propose the following estimate for the $j^{\text{th}}$ coordinate of the mean,

$$\hat{x}_j = \frac{1}{n} \frac{\bar{\beta}}{T(M_j)} \sum_{i=1}^{n} h_{ij}, \tag{4}$$

where $\bar{\beta} \triangleq (\frac{k}{d} \mathbb{E}_{M_j | M_j \geq 1}[\frac{1}{T(M_j)}])^{-1}$. The function $T(M_j)$ changes the scaling of $h_{ij}$ depending on $M_j$, the number of nodes that send the $j^{\text{th}}$ coordinate.

**Lemma 2** (Rand-$k$-Spatial estimator Unbiasedness). *Following the definition of $\bar{\beta}$, we have that the Rand-$k$-Spatial estimator in* (4) *is unbiased, i.e,*

$$\mathbb{E}[\hat{x}_j] = \frac{1}{n} \sum_{i=1}^{n} x_{ij} \tag{5}$$

Observe that the Rand-$k$-estimator of (2) can be obtained by setting $T(M_j) = 1 \ \forall M_j \geq 1$. The estimator described at the beginning of this Section for the case where all the $\mathbf{x}_i$ vectors are identical can be obtained (up to a constant) by setting $T(M_j) = M_j \ \forall M_j \geq 1$ (the constant scaling by $\bar{\beta}$ is required for unbiasedness). In general, the estimators obtained by the different settings of $T(M_j)$ form the Rand-$k$-Spatial family of estimators.

Our goal now is to find the *best* estimator from this family for the task of mean estimation. Towards this end, we first evaluate the mean estimation error for members of this family, and then show how the best choice of $T(M_j)$ is a function of the amount of spatial correlation.

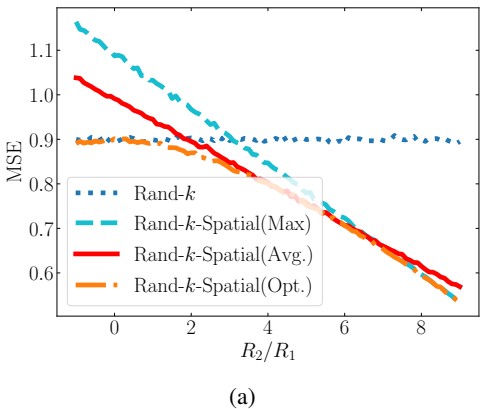 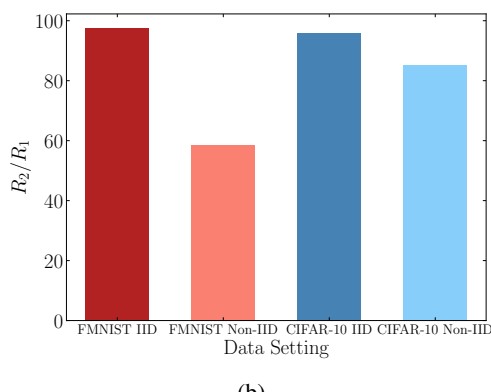

(a)                                  (b)

Figure 2: **(a):** MSE v/s $R_2/R_1$ for $k/d = 0.1$ for the Rand-$k$-Spatial family of estimators. Rand-$k$-Spatial(Avg) significantly outperforms Rand-$k$ and Rand-$k$-Spatial(Max) in the regime where each is sub-optimal (high and low $R_2/R_1$ respectively), while closely tracking the performance of the best (unrealistic) estimator Rand-$k$-Spatial-Opt computed as in (9) with knowledge of $R_2/R_1$. **(b):** $R_2/R_1$ values for a single iteration of distributed Power Iteration on FMNIST and CIFAR datasets (IID & non-IID splits) across 100 nodes. Clearly, $R_2/R_1$ is significantly larger than 0 and thus MSE will *not* be minimized by Rand-$k$.

**Theorem 1** (Rand-$k$-Spatial Estimation Error)**.** *The mean squared error of any member of the Rand-$k$-Spatial family is given by*

$$\mathbb{E}\left[\|\hat{\mathbf{x}} - \bar{\mathbf{x}}\|^2\right] = \frac{1}{n^2}\left(\frac{d}{k} - 1\right)R_1 + \frac{1}{n^2}\left(c_1 R_1 - c_2 R_2\right) \quad (6)$$

*where $R_1 = \sum_{i=1}^{n}\|\mathbf{x}_i\|^2$ and $R_2 = 2\sum_{i}^{n}\sum_{j=i+1}^{n}\langle\mathbf{x}_i,\mathbf{x}_j\rangle$. The parameters $c_1, c_2$ depend on the choice of $T(.)$ as*

$$c_1 = \bar{\beta}^2 \sum_{m=1}^{n} \frac{k}{dT(m)^2}\binom{n-1}{m-1}\left(\frac{k}{d}\right)^{m-1}\left(1 - \frac{k}{d}\right)^{n-m} - \frac{d}{k} \quad (7)$$

*and*

$$c_2 = 1 - \bar{\beta}^2 \sum_{m=2}^{n} \frac{k^2}{d^2 T(m)^2}\binom{n-2}{m-2}\left(\frac{k}{d}\right)^{m-2}\left(1 - \frac{k}{d}\right)^{n-m}. \quad (8)$$

**Choosing the best estimator.** Comparing (6) with (3), we see that if the data $\mathbf{x}_1, \ldots, \mathbf{x}_n$ and function $T()$ is such that $c_1 R_1 < c_2 R_2$ then the MSE of the corresponding estimator in the Rand-$k$-Spatial family is guaranteed to be lower than the Rand-$k$ estimator of Section 2. In general, since the MSE depends on the function $T(.)$ through $c_1$ and $c_2$, we can find the $T(.)$ that minimizes the MSE as shown below.

**Theorem 2** (Rand-$k$-Spatial minimum MSE)**.** *For any set of data vectors $\mathbf{x}_1, \ldots, \mathbf{x}_n$, the ratio $\frac{R_2}{R_1}$ always lies in $[-1, n-1]$, and the estimator within the Rand-$k$ spatial family of estimators that minimizes the MSE in* (6)*, can be obtained by setting*

$$T^*(m) = 1 + \frac{R_2}{R_1}\frac{m-1}{n-1}, \quad for \; m = 1, \ldots, n. \quad (9)$$

Thus, the best mean estimator from the Rand-$k$-family changes as the data (and consequently $R_1$, $R_2$) changes. We now discuss certain special cases of this.

1. **Rand-$k$ (Best when $R_2/R_1 = 0$)** The conventional Rand-$k$ estimator (corresponding to $T^*(m) = 1, \forall m$) minimizes the MSE only in a special case which is, in fact, the case when $R_2 = 2\sum_{i}^{n}\sum_{j=i+1}^{n}\langle\mathbf{x}_i,\mathbf{x}_j\rangle = 0$. A natural setting where this could occur is when the node vectors are orthogonal to each other, i.e., $\langle\mathbf{x}_i,\mathbf{x}_j\rangle = 0 \,\forall i,j \in [n]$. In such cases, it can be said that the conventional Rand-$k$ estimator is, in fact, the best out of the Rand-$k$-Spatial family of estimators. However, when $R_2/R_1 \neq 0$, Rand-$k$ is *not* the best estimator, and for each value of $R_2/R_1$, a different estimator (corresponding to a different value of $T^*(.)$) minimizes the MSE.

2. **Rand-$k$-Spatial(Max) (Best when $R_2/R_1 = n - 1$)** Recall the example described at the beginning of this section where $\mathbf{x}_i = \mathbf{x}_j \; \forall i, j \in [n]$. In this case, $R_2/R_1 = n - 1$, and $T^*(m) = m$, $\forall m$ gives the optimal estimator. We will call this the Rand-$k$-Spatial(Max) estimator since it corresponds to the maximum value of $R_2/R_1$. Thus, the optimal estimators are very different when the vectors are uncorrelated as opposed to when the vectors are highly correlated.

3. **Rand-$k$-Spatial(Avg) (Best on Average)** At this point it is quite clear that no single value of $T(.)$ will minimize MSE for all datasets. However, from (9) it is also evident that to derive the *best* $T(.)$ we need to know the exact values of $R_2/R_1$ at the central server which does not seem possible without communicating the entire vector at each node (particularly for computing $R_2$) which we wish to avoid. Therefore, we propose the following choice of $T(.)$ as a default setting, and subsequently show empirically that it does better than either of the above two extremes on average.

$$\tilde{T}(m) = 1 + \frac{n}{2} \frac{m - 1}{n - 1} \tag{10}$$

Comparing with (9), we see that this is the best estimator when $R_2/R_1 = n/2$ which is the mid-point of the interval in which $R_2/R_1$ lies (as per Theorem 2). It can also be shown that this minimizes the expectation over MSE if $R_2/R_1$ is distributed uniformly over its range. We will now present simulations to show that this estimator gives close to the best MSE across the entire range of $R_2/R_1$, and will show in Section 5 that it can outperform state-of-the-art baselines for real distributed mean estimation tasks.

**Simulations.** To illustrate the discussion above, we performed mean estimation for $n = 10$ $100-$dimensional vectors using $k/d = 0.1$. Since $n = 10$, $R_2/R_1 \in [-1, 9]$ (Theorem 2). We vary $R_2/R_1$ over the entire range by starting with 5 all-1 vectors and 5 all-$(-1)$ vectors ($R_2/R_1 = -1$) and then changing the signs of the elements (one at a time) of the 5 all-$(-1)$ vectors, until they are all-1 ($R_2/R_1 = 9$). For details, see Appendix B. We plot the MSE of the Rand-$k$, Rand-$k$-Spatial(Max), and Rand-$k$-Spatial(Avg) estimators in Fig. 2a. We also add an estimator corresponding to the best $T(.)$ as in (9) and call it Rand-$k$-Spatial(Opt). While knowing $R_2/R_1$ at the central server is unrealistic in practice, it illustrates the best-case scenario. In Fig. 2a, we see that Rand-$k$-Spatial(Avg) is in general quite close to Rand-$k$-Spatial(Opt), i.e., it works well for all values of $R_2/R_1$, whereas Rand-$k$ and Rand-$k$-Spatial(Max) work well only in the special cases where $R_2/R_1 = 0$ and $R_2/R_1 = 9$.

To check if $R_2/R_1$ can indeed take large values for which Rand-$k$ would be suboptimal, we consider distributed power iteration which is a common protocol that uses distributed mean estimation [33, 11]. The objective of the central server here is to estimate the top eigenvector of a matrix whose rows are partitioned among nodes. To simulate this, we partition the FMNIST and CIFAR-10 datasets among 100 nodes, either in an IID or Non-IID fashion and plot $R_2/R_1$ for the vectors in the first round of distributed power iteration on this data. Fig. 2b shows that $R_2/R_1$ is significantly larger than 0 in these cases, thus indicating that the data is well correlated. The values of $R_2/R_1$ continue to remain large even in subsequent round (see Appendix B), thus justifying the need for using estimators other than the conventional Rand-$k$ at each round. In Section 5, we show that the Rand-$k$-Spatial(Avg) estimator can outperform Rand-$k$ and various other baselines for distributed power iteration.

## 4 Improving Random-$k$ by Leveraging Temporal Correlations

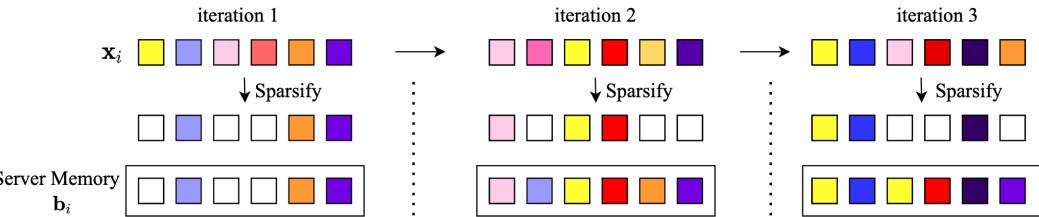

Figure 3: Each node $i$ sends its Rand-$k$ sparsified vector to the server. The server stores previously seen coordinates of the vector in its memory and uses it to fill-in unsampled coordinates in future iterations.

Mean estimation is an important subtask in several iterative algorithms such as power iteration and SGD. We propose a mean estimation technique that leverages temporal correlations in the vectors

$\mathbf{x}_i^{(t)}$ obtained by node $i$, where $t$ is the iteration index of the algorithm. In Rand-$k$ sparsification, each node $i$ samples $k$ out of $d$ coordinates of its vector $\mathbf{x}_i^{(t)}$ to create a sparsified vector $\mathbf{h}_i^{(t)}$ and sends it to the server. In standard Rand-$k$ sparsified mean estimation, the $d-k$ unsampled coordinates of $\mathbf{x}_i^{(t)}$ are treated as 0 by the server when computing the mean estimate (see (2)). Instead, we propose to allow the server to substitute the zeros with values from its memory that *better represent* the unsampled coordinates.

**Filling-in Unsampled Coordinates of a Sparsified Vector.** Let $\mathbf{b}_1^{(t)}, \mathbf{b}_2^{(t)} \dots \mathbf{b}_n^{(t)}$ be $n$ vectors stored in the server memory whose elements serve as substitutes for unsampled coordinates of $\mathbf{x}_i^{(t)}$. Since these $\mathbf{b}_i^{(t)}$s are stored at the server, there is no added computation or storage costs at the nodes. These $\mathbf{b}_i^{(t)}$s can correspond to predicted values of $\mathbf{x}_i^{(t)}$ based on the temporal correlation across iterations. A natural idea is to set $\mathbf{b}_i^{(t)}$ in iteration $t$ to previously sampled coordinates, instead of just substituting zeros. Based on this intuition, we propose the following strategy for updating $\mathbf{b}_i^{(t)}$ at the end of iteration $t$. Let $\mathbf{b}_i^{(0)} = \mathbf{0}$ for all $i \in [n]$. As illustrated in Fig. 3, we update $\mathbf{b}_i^{(t)}$ for $t \geq 1$ as follows:

$$b_{ij}^{(t+1)} \triangleq \begin{cases} b_{ij}^{(t)} & \text{if node } i \text{ does not send } x_{ij}^{(t)} \\ x_{ij}^{(t)} & \text{if node } i \text{ sends } x_{ij}^{(t)}. \end{cases} \tag{11}$$

One can further refine this approach by enforcing staleness constraints on $\mathbf{b}_i^{(t)}$ or by setting $\mathbf{b}_i^{(t)}$ to a sliding window average of previously sampled coordinates.

**The Rand-$k$-Temporal estimator.** We now propose the Rand-$k$-Temporal estimator that enables the server to get an unbiased estimate of the mean $\bar{\mathbf{x}}$ for any arbitrary stored vectors $\mathbf{b}_i$ used to fill in the unsampled coordinates of $\mathbf{x}_i$. We omit the iteration index $t$ here for the purpose of brevity and because this estimator can be used even in non-iterative settings where the server knows a predicted value $\mathbf{b}_i$ of the vector $\mathbf{x}_i$. Given the sparsified vector $\mathbf{h}_i$ received from node $i$, the server can *fill in* the unsampled coordinates and create the vector $\mathbf{h}_i'$ whose $j$-th coordinate is defined as:

$$h_{ij}' \triangleq \begin{cases} b_{ij} & \text{if node } i \text{ does not send } x_{ij} \\ b_{ij} + \frac{d}{k}(x_{ij} - b_{ij}) & \text{if node } i \text{ sends } x_{ij} \end{cases} \tag{12}$$

By definition, $h_{ij}'$ is an unbiased estimate of $x_{ij}$, because $\mathbb{E}\left[h_{ij}'\right] = (1 - \frac{k}{d})b_{ij} + \frac{k}{d}(b_{ij} + \frac{d}{k}(x_{ij} - b_{ij})) = x_{ij}$. Using these filled-in vectors $\mathbf{h}_i'$ obtained from the sparsified vectors $\mathbf{h}_i$ and the vectors $\mathbf{b}_i$ stored in server memory, we propose that the server computes the following mean estimate

$$\hat{\mathbf{x}} = \frac{1}{n} \sum_{i=1}^{n} \mathbf{h}_i'. \tag{13}$$

We see that when $\mathbf{b}_i = 0$ for all $i$, then the Rand-$k$-Temporal estimator reduces to the Rand-$k$ estimator.

Since $\mathbf{h}_i'$ is an unbiased estimate of $\mathbf{x}_i$, it follows that the proposed mean estimate $\hat{\mathbf{x}}$ is also unbiased, that is $\mathbb{E}\left[\hat{\mathbf{x}}\right] = \mathbf{x}$. In the following theorem, we evaluate the estimation error of this Rand-$k$-Temporal estimator in terms of the $\mathbf{b}_i$s.

**Theorem 3** (Rand-$k$-Temporal Estimation Error). *The mean squared error of the Rand-k-Temporal estimator is given by,*

$$\mathbb{E}\left[\|\hat{\mathbf{x}} - \bar{\mathbf{x}}\|^2\right] = \frac{1}{n^2}\left(\frac{d}{k} - 1\right)\sum_{i=1}^{n} \|\mathbf{x}_i - \mathbf{b}_i\|^2. \tag{14}$$

By comparing the above result with the Rand-$k$ estimation error given in Lemma 1, we see that Rand-$k$-Temporal effectively reduces the dependence of the MSE on $\|\mathbf{x}_i\|^2$ to $\|\mathbf{x}_i - \mathbf{b}_i\|^2$. Thus, the Rand-$k$-Temporal estimator can significantly reduce the estimation error of the Rand-$k$ estimator when $\mathbf{b}_i$s are *close* to $\mathbf{x}_i$s, resulting in $\|\mathbf{x}_i - \mathbf{b}_i\|^2 < \|\mathbf{x}_i\|^2$.

**Reducing Storage Cost at the Server.** In the naive implementation of the proposed Rand-$k$-Temporal estimator, we need $\mathcal{O}(nd)$ storage/memory at the server to store vector $\mathbf{b}_i$ from each

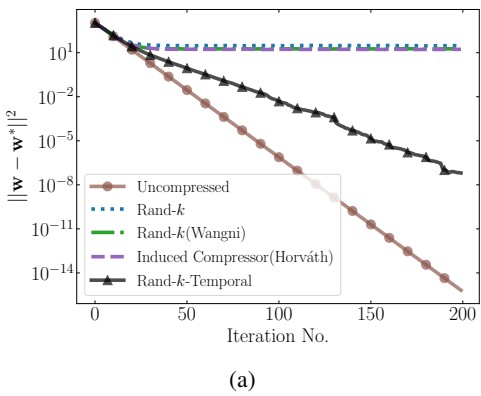
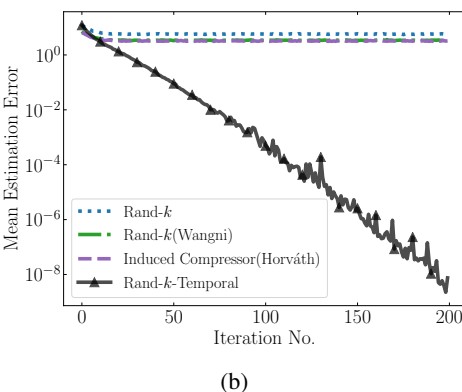

|  (a)  |  (b)  |

Figure 4: Simulated results for our objective defined in (15), where $\mathbf{e}_i \sim \mathcal{N}(0, \mathbf{I}_d)$, $n = 15$ and $d = 1000$. We set $\eta = 0.1$ and $k/d = 0.1$. We compare with more sophisticated schemes than Rand-$k$-Temporal such as Rand-$k$(Wangni) and [36] and Induced Compressor(Horváth) [14] that take into account coordinate magnitude during sparsification. **(a)** We see that Rand-$k$-Temporal effectively converges to the true optima while other sparsification schemes are unable to do so. **(b)** The mean estimation error for Rand-$k$-Temporal decreases sharply as iterations proceed while other sparsification schemes continue to have a high error floor.

device $i$. In practice, this memory cost can be easily reduced at the expense of a slightly higher MSE. Note that our result in Theorem 3 holds for arbitrary $\mathbf{b}_i$ and thus the $\mathbf{b}_i$ do not have to be necessarily different for each device. A natural idea to reduce the memory requirement at the server to just $\mathcal{O}(d)$ is to set $\mathbf{b}_i^{(t+1)} = \hat{\mathbf{x}}^{(t)}$ for all $i \in [n]$, where $\hat{\mathbf{x}}^{(t)}$ is the previous mean estimate. Further details and experimental results confirming the effectiveness of this strategy are shown in Appendix D.

**A Quadratic-Objective Case Study.** Even with the simple choice of $\mathbf{b}_i$ given in (11), Rand-$k$-Temporal estimation gives an *order-wise* improvement in the error over standard Rand-$k$ estimation. We demonstrate this via the following case study of performing gradient descent on a quadratic objective. Consider that each node $i$ has a quadratic local objective function given by $F_i(\mathbf{w}) = \frac{1}{2} \|\mathbf{w} - \mathbf{e}_i\|^2$, where $\mathbf{e}_i \in \mathbb{R}^d$ is the local minimum of node $i$. The local minima are different across nodes due to heterogeneity of their objectives. The goal of the server is to minimize the global objective $F(\mathbf{w})$ given by,

$$F(\mathbf{w}) = \frac{1}{n} \sum_{i=1}^n F_i(\mathbf{w}) = \frac{1}{2n} \sum_{i=1}^n \|\mathbf{w} - \mathbf{e}_i\|^2, \text{ which is minimized by } \mathbf{w}^* = \frac{1}{n} \sum_{i=1}^n \mathbf{e}_i. \quad (15)$$

Our simulation result in Fig. 4b shows how the mean estimation error for these node gradients changes as training progresses. For schemes that do not utilize temporal information, we see a non-zero estimation error due to the unsampled coordinates at all steps. Rand-$k$-Temporal, on the other hand, effectively allows the mean estimation error of the node vectors to converge to zero, by utilizing previous and current information on *all* coordinates while decoding. A consequence of this, as we show in the theorem below, is that Rand-$k$-Temporal converges to the true optima $\mathbf{w}^*$ for a sufficiently small learning rate $\eta$.

**Theorem 4.** *For our objective defined as in* (15)*, Gradient Descent using the Rand-$k$-Temporal estimator with* $\eta \leq \min\left\{ \frac{1}{\left(1 + \frac{8}{n}\left(\frac{d}{k} - 1\right)\right)}, \frac{k}{2d} \right\}$ *converges to the true optima* $\mathbf{w}^*$ *as follows,*

$$\mathbb{E}\left[\left\|\mathbf{w}^{(t+1)} - \mathbf{w}^*\right\|^2\right] \leq (1 - \eta)^{t+1} \left[\left\|\mathbf{w}^0 - \mathbf{w}^*\right\|^2 + \frac{1}{n} \sum_{i=1}^n \|\nabla F_i(\mathbf{w}^*)\|^2\right] \quad (16)$$

*where the expectation is taken over the randomness in the sparsification protocol across all steps.*

**Comparison with Error-Feedback and Gradient Difference methods:** The notion of error-feedback [28, 31] is considered for compressed gradient methods in [29, 30, 20, 6]. In error-feedback, each node locally stores the difference between the actual and the compressed vector at each iteration (known as error), and adds it to the vector generated at the next iteration (feedback). For Rand-$k$

sparsification, error-feedback corresponds to locally storing the unsampled coordinates for each iteration, and adding them back at the next iteration. Note that, with error-feedback, the server still utilizes the information on only $k$ coordinates from each client, albeit with accumulated history. On the other hand, Rand-$k$-Temporal allows the server to leverage temporal correlations by utilizing historical data to substitute for *unsampled* coordinates. Another line of work [26, 15, 22] proposes that nodes quantize the *difference* of their current gradient and a local state to reduce the variance caused by quantization. Note that similar to error-feedback, these works require nodes to maintain a local state and perform extra computation to update the local state. This can incur significant overhead for resource-constrained nodes in many applications such as SGD for deep learning models with millions of parameters.

**Comparison with rTop-k and Induced Compressor:** In practice, biased compressors such as Top-$k$ are observed to perform better due to their low empirical variance, however error-feedback is the only known mechanism to ensure convergence [30, 20, 6, 14]. rTop-k [3] proposes to first find the top $r$ magnitude coordinates of a vector and then randomly sample $k$ out of these $r$ coordinates. Note that rTop-k is still a biased compressor and is usually implemented with error feedback. The work of [14] proposes a general mechanism called induced compressor for constructing an unbiased compression operator from any biased compressor such as Top-$k$. The induced compressor first computes the error between the actual and compressed vector obtained using a biased compressor, compresses the error using an unbiased compressor, and sends it along with the compressed vector. Unlike Rand-$k$-Temporal, the induced compressor only operates on the vectors for the current iteration, and does not leverage temporal correlations.

**Comparison with Lazily Aggregated Gradient (LAG) Methods:** LAG methods [8, 32, 39, 9] seek to utilize temporal correlation to reduce communication. In particular, LAG adaptively skips node-to-server communication by identifying slowly varying gradients, thereby reusing old gradients over time. Unlike Rand-$k$-Temporal, LAG is not a sparsification scheme and incurs an extra computation cost to determine whether or not a node sends their vector to the server.

# 5 Experiments

We evaluate our proposed sparsification schemes on the following three tasks: i) *Power Iteration*, ii) *K-Means*, and iii) *Logistic Regression*. In all setups, we compare the performance with Rand-$k$ and the two more sophisticated sparsification schemes, Rand-$k$ (Wangni) [36], and Induced Compressor [14]. We use the code from [16] for both. For the Induced Compressor, we use Top-$k$ with Rand-$k$ as in [14]. We present representative results here and defer additional results to Appendix C.

**Power Iteration.** We estimate the principal eigenvector of the covariance matrix of the data distributed across 100 nodes by the power iteration method. The data consists of images from the Fashion-MNIST [38] dataset split IID across the nodes. At each iteration, the nodes send a $10\times$ compressed version of their local eigenvector estimates to the server based on a single local power iteration. The server estimates the mean (global eigenvector estimate) and communicates it back to the nodes for the next round. Results in Fig. 5 show that Rand-$k$-Spatial(Avg) (Section 3) and Rand-$k$-Temporal (Section 4) *significantly* outperform *all baselines* both in terms of convergence to the true eigenvector (Fig. 5a) and in terms of the mean estimation error (Fig. 5d) across iterations.

**$K$-Means Clustering.** We perform $K$-Means clustering (with 10 clusters) using Lloyd's algorithm on Fashion-MNIST images distributed across 100 nodes in IID fashion. Each node computes its local cluster centers and sends a $10\times$ compressed version to the server, which estimates the mean (global centers) and sends it back to the nodes for the next round. Results in Fig. 5b and Fig. 5e show that Rand-$k$-Spatial(Avg) (Section 3) and Rand-$k$-Temporal (Section 4) *significantly* outperform *all baselines* both in terms of minimizing the $K$-Means objective (the average distance to cluster centers) and in terms of the average mean estimation error averaged over the 10 cluster means.

**Logistic Regression.** We train a logistic regression model for classification of CIFAR-10 images split across 10 nodes in non-IID fashion (following the splitting procedure in [25]). Each node computes local gradients and sends a $100\times$ compressed version to the server, which estimates the mean (global gradient) and sends it back to the nodes for the next round. Results show that Rand-$k$-Temporal achieves the lowest training loss (Fig. 5c) and order-wise lower mean estimation error (Fig. 5f) among all sparsification schemes as training progresses. We note that Rand-$k$-Spatial is not able to beat the

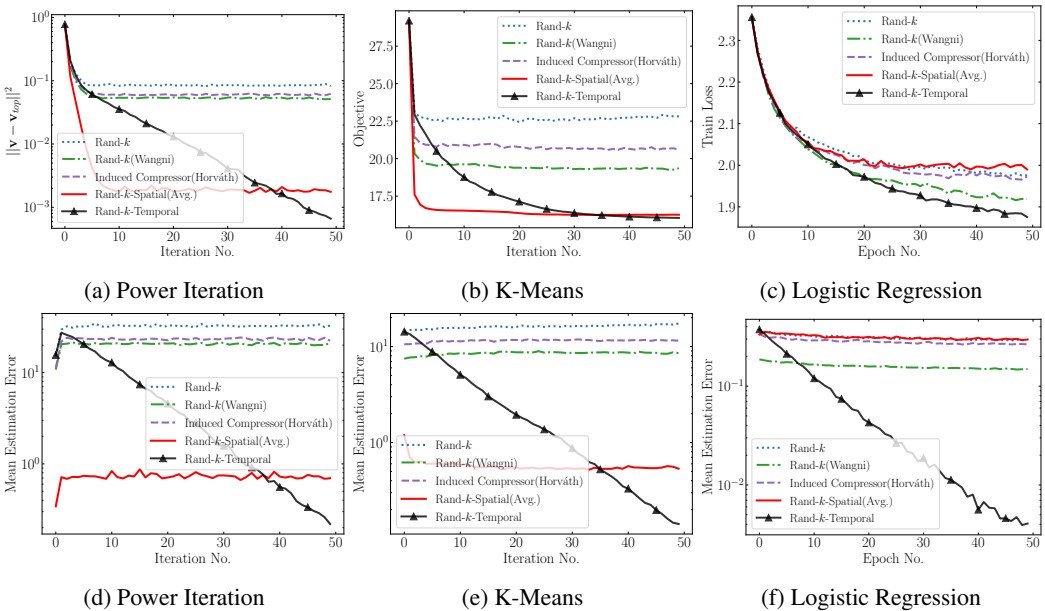

Figure 5: Experimental Results showing task objective and Mean Estimation Error for Power Iteration (a,d), K-Means (c,e), and Logistic Regression (c,f). Rand-$k$-Temporal outperforms baselines in all cases while Rand-$k$-Spatial outperforms baselines in 2/3 cases.

baselines here potentially because the non-IID split might be reducing spatial correlation. We plan to investigate this further in future work.

## 6   Conclusion

We considered the problem of estimating the mean of $d$-dimensional vectors received from $n$ nodes, which sparsify their vectors to $k$ elements in order to reduce the communication cost. When there are spatial correlations across nodes and temporal correlations over time between the vectors, the standard approach of simple averaging the sparsified vectors can lead to high estimation error. We propose the Rand-$k$-Spatial and Rand-$k$-Temporal estimators that provably reduce the estimation error while keeping the estimate unbiased and also perform exceptionally well in experiments. Although we focus on Rand-$k$ sparsification here, the insights for leveraging correlations can be extended to other sparsification and compression methods.

## Acknowledgments

This research was generously supported in part by the NSF Award (CNS-2112471), the NSF CRII Award (CCF-1850029), the NSF CAREER Award (CCF-2045694), the NSF MLWiNS Award (MLWiNS-2002821), the Qualcomm Innovation fellowship and the CMU Dean's fellowship.

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
