# A  Proof of Theoretical Results

## A.1  Proof of Lemma 1

The MSE can be computed as

$$\mathbb{E}\left[\|\hat{\mathbf{x}} - \bar{\mathbf{x}}\|^2\right] = \sum_{j=1}^{d} \mathbb{E}\left[(\hat{x}_j - \bar{x}_j)^2\right] = \sum_{j=1}^{d} \mathbb{E}\left[\left(\frac{1}{n}\sum_{i=1}^{n}\frac{d}{k}h_{ij} - \frac{1}{n}\sum_{i=1}^{n}x_{ij}\right)^2\right]. \tag{17}$$

Now, as $\mathbb{E}\left[\frac{d}{k}h_{ij}\right] = x_{ij}$, it holds that

$$\frac{1}{n^2}\sum_{i=1}^{n}\mathbb{E}\left[\left(\frac{d}{k}h_{ij} - x_{ij}\right)^2\right] = \frac{1}{n^2}\sum_{i=1}^{n}\left(\mathbb{E}\left[\left(\frac{d}{k}h_{ij}\right)^2\right] - x_{ij}^2\right). \tag{18}$$

Since $h_{ij} = x_{ij}$ with probability $k/d$ and $h_{ij} = 0$ otherwise (by definition), therefore $\mathbb{E}\left[\left(\frac{d}{k}h_{ij}\right)^2\right] = \frac{d}{k}x_{ij}^2$, which implies

$$\mathbb{E}\left[\|\hat{\mathbf{x}} - \bar{\mathbf{x}}\|^2\right] = \frac{1}{n^2}\left(\frac{d}{k} - 1\right)\sum_{i=1}^{n}\sum_{j=1}^{d}x_{ij}^2 = \frac{1}{n^2}\left(\frac{d}{k} - 1\right)R_1, \tag{19}$$

where $R_1 = \sum_{i=1}^{n}\|\mathbf{x}_i\|^2$.

## A.2  Proof of Lemma 2

We begin by recalling our definition of $\bar{\beta}$.

$$\bar{\beta} \triangleq \left(\frac{k}{d}\mathbb{E}_{M_j|M_j \geq 1}\left[\frac{1}{T(M_j)}\right]\right)^{-1} \tag{20}$$

Let $\xi_{ij}$ be an indicator random variable which is 1 or 0, depending on whether $h_{ij} = x_{ij}$ or not.

**Case 1:** With probability $(1 - \frac{k}{d})$, $\xi_{ij} = 0$ which implies $h_{ij} = 0$. Therefore,

$$\mathbb{E}_{M_j|\xi_{ij}=0}\left[\frac{\bar{\beta}h_{ij}}{T(M_j)}\right] = 0 \tag{21}$$

**Case 2:** With probability $\frac{k}{d}$, $\xi_{ij} = 1$ which implies $h_{ij} = x_{ij}$. Therefore,

$$\mathbb{E}_{M_j|\xi_{ij}=1}\left[\frac{\bar{\beta}h_{ij}}{T(M_j)}\right] = \mathbb{E}_{M_j|M_j \geq 1}\left[\frac{\bar{\beta}h_{ij}}{T(M_j)}\right] = \bar{\beta}x_{ij}\mathbb{E}_{M_j|M_j \geq 1}\left[\frac{1}{T(M_j)}\right] \tag{22}$$

The crucial observation here is that $\xi_{ij} = 1$ only implies $M_j \geq 1$ and does not give any other information about $M_j$. This allows us to decouple the relation between $h_{ij}$ and $M_j$ for this proof as well as our proof of Theorem 1. Next, taking expectation with respect to $\xi_{ij}$ we have,

$$\mathbb{E}_{\xi_{ij}}\mathbb{E}_{M_j|\xi_{ij}} = \frac{k}{d}\bar{\beta}x_{ij}\mathbb{E}_{M_j|M_j \geq 1}\left[\frac{1}{T(M_j)}\right] = x_{ij} \tag{23}$$

which follows from the definition of $\bar{\beta}$. This proves that the estimate (4) is unbiased.

## A.3  Proof of Theorem 1

The MSE can be computed as

$$\mathbb{E}\left[\|\hat{\mathbf{x}} - \bar{\mathbf{x}}\|^2\right] = \sum_{j=1}^{d}\mathbb{E}\left[(\hat{x}_j - \bar{x}_j)^2\right] = \sum_{j=1}^{d}\mathbb{E}\left[\left(\frac{1}{n}\frac{\bar{\beta}}{T(M_j)}\sum_{i=1}^{n}h_{ij} - \frac{1}{n}\sum_{i=1}^{n}x_{ij}\right)^2\right]. \tag{24}$$

As the estimator is designed to be unbiased, i.e., $\mathbb{E}\left[\left(\frac{1}{n}\frac{\bar{\beta}}{T(M_j)}\sum_{i=1}^{n}h_{ij}\right)\right]=\frac{1}{n}\sum_{i=1}^{n}x_{ij}$, it holds that

$$\mathbb{E}\left[\left(\frac{1}{n}\frac{\bar{\beta}}{T(M_j)}\sum_{i=1}^{n}h_{ij}-\frac{1}{n}\sum_{i=1}^{n}x_{ij}\right)^2\right]=\frac{1}{n^2}\mathbb{E}\left[\left(\frac{\bar{\beta}}{T(M_j)}\sum_{i=1}^{n}h_{ij}\right)^2\right]-\frac{1}{n^2}\left(\sum_{i=1}^{n}x_{ij}\right)^2 \quad (25)$$

We now analyze the first term above.

$$\mathbb{E}\left[\left(\frac{\bar{\beta}}{T(M_j)}\sum_{i=1}^{n}h_{ij}\right)^2\right]=\sum_{i=1}^{n}\bar{\beta}^2\mathbb{E}\left[\frac{h_{ij}}{T(M_j)}\right]^2+\sum_{i=1}^{n}\sum_{k=i+1}^{n}\bar{\beta}^2\mathbb{E}\left[\frac{h_{ij}h_{kj}}{T(M_j)^2}\right] \quad (26)$$

Note here that the expectation is taken over the randomness in $h_{ij}$ as well as $T(M_j)$. Further, $\bar{\beta}^2\left[\frac{h_{ij}}{T(M_j)}\right]^2$ is non-zero only when a node $i$ samples coordinate $j$, i.e., $h_{ij}=x_{ij}$. This implies that $M_j\geq 1$. Therefore, by the law of total expectation, we have

$$\bar{\beta}^2\mathbb{E}\left[\frac{h_{ij}}{T(M_j)}\right]^2=\bar{\beta}^2\mathbb{E}_{M_j|M_j\geq 1}\left[\frac{kx_{ij}^2}{dT(M_j)^2}\right] \quad (27)$$

$$=\left(\bar{\beta}^2\sum_{m=1}^{n}\frac{k}{dT(m)^2}\binom{n-1}{m-1}\left(\frac{k}{d}\right)^{m-1}\left(1-\frac{k}{d}\right)^{n-m}\right)x_{ij}^2 \quad (28)$$

$$=\left(\frac{d}{k}+c_1\right)x_{ij}^2 \quad (29)$$

where $c_1=\bar{\beta}^2\sum_{m=1}^{n}\frac{k}{dT(m)^2}\binom{n-1}{m-1}\left(\frac{k}{d}\right)^{m-1}\left(1-\frac{k}{d}\right)^{n-m}-\frac{d}{k}$. Here, the second equality uses the fact that when node $i$ samples coordinate $j$ (i.e., $x_{ij}=h_{ij}$), then $M_j\geq 1$.

Following a similar argument as above, note that $\bar{\beta}^2\left[\frac{h_{ij}h_{kj}}{T(M_j)^2}\right]$ is non-zero only when nodes $i$ and $k$ sample coordinate $j$, i.e., $h_{ij}=x_{ij}$ and $h_{kj}=x_{kj}$. This implies that $M_j\geq 2$. Therefore, by the law of total expectation, we have

$$\bar{\beta}^2\mathbb{E}\left[\frac{h_{ij}h_{kj}}{T(M_j)^2}\right]=\bar{\beta}^2\mathbb{E}_{M_j|M_j\geq 2}\left[\frac{k^2x_{ij}x_{kj}^2}{d^2T(M_j)^2}\right] \quad (30)$$

$$=\left(\bar{\beta}^2\sum_{m=2}^{n}\frac{k^2}{d^2T(m)^2}\binom{n-2}{m-2}\left(\frac{k}{d}\right)^{m-2}\left(1-\frac{k}{d}\right)^{n-m}\right)x_{ij}x_{kj} \quad (31)$$

$$=(1-c_2)x_{ij}x_{kj}, \quad (32)$$

where $c_2=1-\bar{\beta}^2\sum_{m=2}^{n}\frac{k^2}{d^2T(m)^2}\binom{n-2}{m-2}\left(\frac{k}{d}\right)^{m-2}\left(1-\frac{k}{d}\right)^{n-m}$

Substituting (29) and (32) in (26), we get

$$\mathbb{E}\left[\left(\frac{\bar{\beta}}{T(M_j)}\sum_{i=1}^{n}h_{ij}\right)^2\right]=\left(\frac{d}{k}+c_1\right)\sum_{i=1}^{n}x_{ij}^2+(1-c_2)\sum_{i=1}^{n}\sum_{k=i+1}^{n}x_{ij}x_{kj} \quad (33)$$

Now, substituting (33) in (25), we get

$$\mathbb{E}\left[\left(\frac{1}{n}\frac{\bar{\beta}}{T(M_j)}\sum_{i=1}^{n}h_{ij}-\frac{1}{n}\sum_{i=1}^{n}x_{ij}\right)^2\right]=\frac{1}{n^2}\left(\frac{d}{k}+c_1-1\right)\sum_{i=1}^{n}x_{ij}^2-\frac{1}{n^2}c_2\sum_{i=1}^{n}\sum_{k=i+1}^{n}x_{ij}x_{kj} \quad (34)$$

Finally replacing (34) in (24) we get,

$$\mathbb{E}\left[\|\hat{\mathbf{x}}-\bar{\mathbf{x}}\|^2\right]=\frac{1}{n^2}\left(\frac{d}{k}-1\right)R_1+\frac{1}{n^2}\left(c_1R_1-c_2R_2\right) \quad (35)$$

where $R_1=\sum_{i=1}^{n}\|\mathbf{x}_i\|^2$ and $R_2=2\sum_{i}^{n}\sum_{j=i+1}^{n}\langle\mathbf{x}_i,\mathbf{x}_j\rangle$.

## A.4 Proof of Theorem 2

Observe that in (6), the only term that depends on $T(.)$ is $c_1 R_1 - c_2 R_2$. Thus to find the function $T^*(.)$ that minimizes the MSE, we just need to minimize this term.

Recall that $R_1 = \sum_{i=1}^{n} \|\mathbf{x}_i\|^2$ and $R_2 = 2 \sum_i^n \sum_{j=i+1}^{n} \langle \mathbf{x}_i, \mathbf{x}_j \rangle$. Note that $R_1 + R_2 = \|\sum_{i=1}^{n} \mathbf{x}_i\|^2$. Since $\|\sum_{i=1}^{n} \mathbf{x}_i\|^2 \geq 0$ and $\|\sum_{i=1}^{n} \mathbf{x}_i\|^2 \leq n R_1$, it follows that $\frac{R_2}{R_1} \in [-1, n-1]$.

Next, from the definitions of $c_1$ and $c_2$ in (7) and (8) respectively, we can obtain the following expression for $T^*$

$$T^*(m) = \arg\min_T \bar{\beta}^2 \sum_{m=1}^{n} \frac{k}{dT(m)^2} \binom{n-1}{m-1} \left(\frac{k}{d}\right)^{m-1} \left(1 - \frac{k}{d}\right)^{n-m}$$
$$+ \frac{R_2}{R_1} \bar{\beta}^2 \sum_{m=2}^{n} \frac{k^2}{d^2 T(m)^2} \binom{n-2}{m-2} \left(\frac{k}{d}\right)^{m-2} \left(1 - \frac{k}{d}\right)^{n-m}, \tag{36}$$

where $\bar{\beta} = \left( \sum_{m=1}^{n} \frac{k}{dT(m)} \binom{n-1}{m-1} \left(\frac{k}{d}\right)^{m-1} \left(1 - \frac{k}{d}\right)^{n-m} \right)^{-1}$.

We claim that $T^*(m) = 1 + \frac{R_2}{R_1} \frac{m-1}{n-1}$ is an optimal solution for our objective defined in (36). To see this, consider the following cases,

**Case 1:** $p = 0$ or $p = 1$.

In this case $c_1$ and $c_2$ are independent of $T(.)$ and hence our objective does not depend on the choice of $T(.)$.

**Case 2:** $0 < p < 1$ and $\frac{R_2}{R_1} = -1$.

Since $\frac{R_2}{R_1} = -1$ this implies $\|\sum_{i=1}^{n} \mathbf{x}_i\|^2 = 0$ and therefore $\bar{\mathbf{x}} = \mathbf{0}$.

Note that in this case $T^*(n) = 0 \implies \bar{\beta} = 0 \implies \hat{\mathbf{x}} = \mathbf{0}$, thereby recovering the true mean with zero MSE.

**Case 3:** $0 < p < 1$ and $\frac{R_2}{R_1} \in (-1, n-1]$

We define

$$\mathbf{w}^* = \arg\min_{\mathbf{w}} \frac{\mathbf{w}^T \mathbf{A} \mathbf{w}}{(\mathbf{b}^T \mathbf{w})^2}, \tag{37}$$

where $\mathbf{w}$ is a $n$-dimensional vector whose $m^{th}$ entry is $w_m = 1/T(m)$, $\mathbf{b}$ is a vector whose $m^{th}$ entry is

$$b_m = \binom{n-1}{m-1} p^{m-1} (1-p)^{n-m}, \tag{38}$$

where $p = k/d$, and $\mathbf{A}$ is a diagonal matrix whose $m^{th}$ diagonal entry is

$$A_{mm} = \binom{n-1}{m-1} p^{m-1} (1-p)^{n-m} + p * \frac{R_2}{R_1} \binom{n-2}{m-2} p^{m-2} (1-p)^{n-m} \tag{39}$$

$$= b_m (1 + \frac{R_2}{R_1} \frac{m-1}{n-1}). \tag{40}$$

Note that $A_{mm} > 0$ for all $m \in \{1, 2, \ldots, n\}$ which implies that $\mathbf{w} \to \mathbf{A}^{1/2} \mathbf{w}$ is a one-to-one mapping. Therefore setting $\mathbf{z} = \mathbf{A}^{1/2} \mathbf{w}$, the objective in (37) reduces to

$$\mathbf{z}^* = \arg\min_{\mathbf{z}} \frac{\|\mathbf{z}\|^2}{(\mathbf{b}^T \mathbf{A}^{-1/2} \mathbf{z})^2} \tag{41}$$

Observe that the objectives (36), (37), (41) are invariant to the scale of $T(.)$, $\mathbf{w}$, and $\mathbf{z}$ respectively and thus the solutions will be unique up to a scaling factor (this doesn't affect our estimate of the

mean in (4) since $\bar{\beta}$ will be adjusted accordingly). Therefore, in the case of (41), it is sufficient to solve the reduced objective,

$$\mathbf{z}^* = \arg\min_{\mathbf{z},\ ||\mathbf{z}||=1} \frac{||\mathbf{z}||^2}{(\mathbf{b}^T \mathbf{A}^{-1/2}\mathbf{z})^2} = \arg\min_{\mathbf{z},\ ||\mathbf{z}||=1} \frac{1}{(\mathbf{b}^T \mathbf{A}^{-1/2}\mathbf{z})^2} \tag{42}$$

which is minimized (denominator is maximized) by $\mathbf{z}^* = \frac{\mathbf{A}^{-1/2}\mathbf{b}}{||\mathbf{A}^{-1/2}\mathbf{b}||}$. Therefore, the optimal solution (up to a constant) is $\mathbf{w}^* = \mathbf{A}^{-1/2}(\mathbf{A}^{-1/2}\mathbf{b})$. Correspondingly, we have that

$$T^*(m) = \frac{1}{w_m^*} = \frac{A_{mm}}{b_m} = 1 + \frac{R_2}{R_1}\frac{m-1}{n-1}. \tag{43}$$

minimizes (36), and consequently minimizes the MSE of the Rand-$k$-Spatial family of estimators.

### A.5   Proof of Theorem 3

The MSE can be written as

$$\mathbb{E}\left[||\hat{\mathbf{x}} - \bar{\mathbf{x}}||^2\right] = \sum_{j=1}^{d} \mathbb{E}\left[(\hat{x}_j - \bar{x}_j)^2\right] = \sum_{j=1}^{d} \mathbb{E}\left[\left(\frac{1}{n}\sum_{i=1}^{n} h'_{ij} - \frac{1}{n}\sum_{i=1}^{n} x_{ij}\right)^2\right]. \tag{44}$$

Since $\mathbb{E}\left[h'_{ij}\right] = x_{ij}$, it holds that

$$\frac{1}{n^2}\sum_{i=1}^{n} \mathbb{E}\left[(h'_{ij} - x_{ij})^2\right] = \frac{1}{n^2}\sum_{i=1}^{n}\left(\mathbb{E}\left[(h'_{ij})^2\right] - x_{ij}^2\right). \tag{45}$$

From the definition of $h'_{ij}$, we see that

$$\mathbb{E}\left[(h'_{ij})^2\right] - x_{ij}^2 = \left(1 - \frac{k}{d}\right)b_{ij}^2 + \frac{k}{d}\left(b_{ij}^2 + \frac{d^2}{k^2}(x_{ij} - b_{ij})^2 + \frac{2d}{k}b_{ij}(x_{ij} - b_{ij})\right) - x_{ij}^2 \tag{46}$$

$$= \frac{d}{k}(x_{ij} - b_{ij})^2 - (x_{ij} - b_{ij})^2 \tag{47}$$

$$= \left(\frac{d}{k} - 1\right)(x_{ij} - b_{ij})^2. \tag{48}$$

This implies,

$$\mathbb{E}\left[||\hat{\mathbf{x}} - \bar{\mathbf{x}}||^2\right] = \frac{1}{n^2}\left(\frac{d}{k} - 1\right)\sum_{i=1}^{n}\sum_{j=1}^{d}(x_{ij} - b_{ij})^2 = \frac{1}{n^2}\left(\frac{d}{k} - 1\right)\sum_{i=1}^{n}||\mathbf{x}_i - \mathbf{b}_i||^2 \tag{49}$$

### A.6   Proof of Theorem 4

At round $t$, nodes receive the current global model $\mathbf{w}^{(t)}$ from the server and calculate their gradients given by,

$$\mathbf{x}_i^{(t)} = \nabla F_i(\mathbf{w}^{(t)}) \tag{50}$$

$$= \mathbf{w}^{(t)} - \mathbf{e}_i \tag{51}$$

which are then sparsified and averaged using the Rand-$k$-Temporal estimator to update $\mathbf{w}^{(t)}$ as follows,

$$\mathbf{w}^{(t+1)} = \mathbf{w}^{(t)} - \eta\hat{\mathbf{x}}. \tag{52}$$

$$= \mathbf{w}^{(t)} - \eta\frac{1}{n}\sum_{i=1}^{n}\mathbf{h}_i'^{(t)} \tag{53}$$

Our goal is to bound $\mathbb{E}\left[\left\|\mathbf{w}^{(t+1)} - \mathbf{w}^*\right\|^2\right]$. We first define the following quantities that we use in our proof.

$$p \triangleq \frac{k}{d}; \qquad \alpha \triangleq \frac{1}{n}\left(\frac{d}{k} - 1\right); \qquad y^{(t)} \triangleq \frac{1}{n}\sum_{i=1}^{n}\mathbb{E}\left[\left\|\mathbf{b}_i^{(t)} - \nabla F_i(\mathbf{w}^*)\right\|^2\right] \tag{54}$$

Let $\xi^{(t)}$ denote the randomness resulting due to the random sparsification at nodes at round $t$. We have,

$$\mathbb{E}_{\xi^{(t)}}\left[\left\|\mathbf{w}^{(t+1)} - \mathbf{w}^*\right\|^2\right]$$

$$= \mathbb{E}_{\xi^{(t)}}\left[\left\|\mathbf{w}^{(t)} - \eta\hat{\mathbf{x}}^{(t)} - \mathbf{w}^*\right\|^2\right] \tag{55}$$

$$= \left\|\mathbf{w}^{(t)} - \mathbf{w}^*\right\|^2 - 2\eta\mathbb{E}_{\xi^{(t)}}\left[\langle\hat{\mathbf{x}}^{(t)}, \mathbf{w}^{(t)} - \mathbf{w}^*\rangle\right] + \eta^2\mathbb{E}_{\xi^{(t)}}\left[\left\|\hat{\mathbf{x}}^{(t)}\right\|^2\right] \tag{56}$$

$$= \left\|\mathbf{w}^{(t)} - \mathbf{w}^*\right\|^2 - 2\eta\langle\nabla F(\mathbf{w}^{(t)}), \mathbf{w}^{(t)} - \mathbf{w}^*\rangle + \eta^2\mathbb{E}_{\xi^{(t)}}\left[\left\|\hat{\mathbf{x}}^{(t)}\right\|^2\right]$$
$$\left(\text{ since } \mathbb{E}_{\xi^{(t)}}\left[\hat{\mathbf{x}}^{(t)}\right] = \nabla F(\mathbf{w}^{(t)})\right) \tag{57}$$

$$= \left\|\mathbf{w}^{(t)} - \mathbf{w}^*\right\|^2 - 2\eta\langle\nabla F(\mathbf{w}^{(t)}), \mathbf{w}^{(t)} - \mathbf{w}^*\rangle + \eta^2\left\|\nabla F(\mathbf{w}^{(t)})\right\|^2$$
$$+ \eta^2\frac{1}{n^2}\left(\frac{d}{k} - 1\right)\sum_{i=1}^{n}\left\|\nabla F_i(\mathbf{w}^{(t)}) - \mathbf{b}_i^{(t)}\right\|^2 \qquad (\text{ using bias-variance decomposition}) \tag{58}$$

$$= (1 - \eta)^2\left\|\mathbf{w}^{(t)} - \mathbf{w}^*\right\|^2 + \eta^2\alpha\frac{1}{n}\sum_{i=1}^{n}\left\|\nabla F_i(\mathbf{w}^{(t)}) - \mathbf{b}_i^{(t)}\right\|^2 \,(\text{ since } \nabla F(\mathbf{w}^{(t)}) = \mathbf{w}^{(t)} - \mathbf{w}^*)$$
$$\tag{59}$$

$$\leq (1 - \eta)^2\left\|\mathbf{w}^{(t)} - \mathbf{w}^*\right\|^2 + 2\eta^2\alpha\frac{1}{n}\sum_{i=1}^{n}\left\|\nabla F_i(\mathbf{w}^{(t)}) - \nabla F_i(\mathbf{w}^*)\right\|^2 +$$

$$+ 2\eta^2\alpha\frac{1}{n}\sum_{i=1}^{n}\left\|\mathbf{b}_i^{(t)} - \nabla F_i(\mathbf{w}^*)\right\|^2 \qquad (\text{ using } \|a + b\|^2 \leq 2\|a\|^2 + 2\|b\|^2) \tag{60}$$

$$= ((1 - \eta)^2 + 2\eta^2\alpha)\left\|\mathbf{w}^{(t)} - \mathbf{w}^*\right\|^2 + 2\eta^2\alpha\frac{1}{n}\sum_{i=1}^{n}\left\|\mathbf{b}_i^{(t)} - \nabla F_i(\mathbf{w}^*)\right\|^2 \tag{61}$$

Recall the update rule of $b_{ij}^{(t+1)}$.

$$b_{ij}^{(t+1)} = \begin{cases} b_{ij}^{(t)} & \text{with probability } 1 - p \\ \left(\nabla F_i(\mathbf{w}^{(t)})\right)_j & \text{with probability } p \end{cases} \tag{62}$$

This gives us,

$$\mathbb{E}_{\xi^{(t)}}\left[\left\|\mathbf{b}_i^{(t+1)} - \nabla F_i(\mathbf{w}^*)\right\|^2\right] = \sum_{j=1}^{d} \mathbb{E}_{\xi^{(t)}}\left[\left\|b_{ij}^{(t+1)} - (\nabla F_i(\mathbf{w}^*))_j\right\|^2\right] \tag{63}$$

$$= \sum_{j=1}^{d}\left(p\left\|\left(\nabla F_i(\mathbf{w}^{(t)})\right)_j - (\nabla F_i(\mathbf{w}^*))_j\right\|^2\right.$$

$$\left. + (1-p)\left\|b_{ij}^{(t)} - (\nabla F_i(\mathbf{w}^*))_j\right\|^2\right) \tag{64}$$

$$= p\left\|\nabla F_i(\mathbf{w}^{(t)}) - \nabla F_i(\mathbf{w}^*)\right\|^2 + (1-p)\left\|\mathbf{b}_i^{(t)} - \nabla F_i(\mathbf{w}^*)\right\|^2 \tag{65}$$

$$= p\left\|\mathbf{w}^{(t)} - \mathbf{w}^*\right\|^2 + (1-p)\left\|\mathbf{b}_i^{(t)} - \nabla F_i(\mathbf{w}^*)\right\|^2 \tag{66}$$

Therefore,

$$\frac{1}{n}\sum_{i=1}^{n}\mathbb{E}_{\xi^{(t)}}\left[\left\|\mathbf{b}_i^{(t+1)} - \nabla F_i(\mathbf{w}^*)\right\|^2\right] = p\left\|\mathbf{w}^{(t)} - \mathbf{w}^*\right\|^2 + (1-p)\frac{1}{n}\sum_{i=1}^{n}\left\|\mathbf{b}_i^{(t)} - \nabla F_i(\mathbf{w}^*)\right\|^2 \tag{67}$$

This implies,

$$y^{(t+1)} = p\mathbb{E}\left[\left\|\mathbf{w}^{(t)} - \mathbf{w}^*\right\|^2\right] + (1-p)y^{(t)} \tag{68}$$

Using (61) we have,

$$\mathbb{E}\left[\left\|\mathbf{w}^{(t+1)} - \mathbf{w}^*\right\|^2\right] \le \left((1-\eta)^2 + 2\eta^2\alpha\right)\mathbb{E}\left[\left\|\mathbf{w}^{(t)} - \mathbf{w}^*\right\|^2\right] + 2\eta^2\alpha y^{(t)} \tag{69}$$

Using (68) we can unroll the dependence on $y^{(t)}$ to get,

$$\mathbb{E}\left[\left\|\mathbf{w}^{(t+1)} - \mathbf{w}^*\right\|^2\right] \le \left((1-\eta)^2 + 2\eta^2\alpha\right)\mathbb{E}\left[\left\|\mathbf{w}^{(t)} - \mathbf{w}^*\right\|^2\right]$$

$$+ 2\eta^2\alpha p\sum_{k=0}^{t-1}(1-p)^k\mathbb{E}\left[\left\|\mathbf{w}^{(t-1-k)} - \mathbf{w}^*\right\|^2\right] + 2\eta^2\alpha(1-p)^t y^0 \tag{70}$$

We define the quantity $G$ as follows,

$$G \triangleq \left\|\mathbf{w}^0 - \mathbf{w}^*\right\|^2 + y^0 \tag{71}$$

$$= \left\|\mathbf{w}^0 - \mathbf{w}^*\right\|^2 + \frac{1}{n}\sum_{i=1}^{n}\left\|\nabla F_i(\mathbf{w}^*)\right\|^2 \tag{72}$$

We now claim that $\mathbb{E}\left[\left\|\mathbf{w}^{(t+1)} - \mathbf{w}^*\right\|^2\right] \le (1-\eta)^{t+1}G$ for a sufficiently small value of $\eta$.

To show this we use the following inductive argument- assume that $\mathbb{E}\left[\left\|\mathbf{w}^{(k)} - \mathbf{w}^*\right\|^2\right] \le (1-\eta)^k G$ holds for all $k \le t$, then $\mathbb{E}\left[\left\|\mathbf{w}^{(t+1)} - \mathbf{w}^*\right\|^2\right] \le (1-\eta)^{t+1}G$. Note that $\left\|\mathbf{w}^{(0)} - \mathbf{w}^*\right\|^2 \le G$ is already satisfied along with $y^0 \le G$, by the definition of $G$.

Now using our result in (70) and our inductive assumption we have,

$$\mathbb{E}\left[\left\|\mathbf{w}^{(t+1)} - \mathbf{w}^*\right\|^2\right] \leq \left((1-\eta)^2 + 2\eta^2\alpha\right)(1-\eta)^t G$$

$$+ 2\eta^2\alpha p \left(\sum_{k=0}^{t-1}(1-p)^k(1-\eta)^{t-1-k}\right)G + 2\eta^2\alpha(1-p)^t G \qquad (73)$$

$$= (1-\eta)^t\left[\left((1-\eta)^2 + 2\eta^2\alpha\right)\right.$$

$$\left. + \frac{2\eta^2\alpha p}{1-\eta}\left(\sum_{k=0}^{t-1}\left[\frac{1-p}{1-\eta}\right]^k\right) + 2\eta^2\alpha\left[\frac{1-p}{1-\eta}\right]^t\right]G \qquad (74)$$

$$\leq (1-\eta)^t\left[\left((1-\eta)^2 + 2\eta^2\alpha\right) + \frac{2\eta^2\alpha p}{p-\eta} + 2\eta^2\alpha\right]G \qquad (75)$$

where the last line follows from the condition that $\eta < p$. Our goal is to now find a condition on $\eta$ such that,

$$(1-\eta)^2 + 2\eta^2\alpha + \frac{2\eta^2\alpha p}{p-\eta} + 2\eta^2\alpha \leq (1-\eta) \qquad (76)$$

We first impose the condition that $\frac{p}{p-\eta} \leq 2$. This gives us

$$\eta \leq \frac{p}{2} \qquad (77)$$

Now to satisfy,

$$(1-\eta)^2 + 8\eta^2\alpha \leq 1 - \eta \qquad (78)$$

we must have,

$$\eta \leq \frac{1}{1+8\alpha} \qquad (79)$$

Therefore for $\eta \leq \min\left\{\frac{1}{1+8\alpha}, \frac{p}{2}\right\}$ we have,

$$\mathbb{E}\left[\left\|\mathbf{w}^{(t+1)} - \mathbf{w}^*\right\|^2\right] \leq (1-\eta)^{t+1}\left[\left\|\mathbf{w}^0 - \mathbf{w}^*\right\|^2 + \frac{1}{n}\sum_{i=1}^{n}\|\nabla F_i(\mathbf{w}^*)\|^2\right] \qquad (80)$$

thereby completing our proof.

Note that our proof can be easily extended to the case where $F(\mathbf{w})$ is $\mu$-strongly convex and $F_i(\mathbf{w})$ are $L$-smooth functions by using the appropriate convexity and smoothness properties in equations (59) and (66) respectively.

# B Simulation Results:

## B.1 MSE vs $R_2/R_1$ for Rand-$k$-Spatial family of estimators

We first note that,

$$\frac{R_2}{R_1} = \frac{2\sum_i^n \sum_{j=i+1}^n \langle \mathbf{x}_i, \mathbf{x}_j \rangle}{\sum_{i=1}^n \|\mathbf{x}_i\|^2} \tag{81}$$

$$= \frac{\|\sum_{i=1}^n \mathbf{x}_i\|^2 - \sum_{i=1}^n \|\mathbf{x}_i\|^2}{\sum_{i=1}^n \|\mathbf{x}_i\|^2} \tag{82}$$

$$= \frac{\|\sum_{i=1}^n \mathbf{x}_i\|^2}{\sum_{i=1}^n \|\mathbf{x}_i\|^2} - 1 \tag{83}$$

For the purpose of this simulation, we first generate 5 all-(1) vectors and 5 all-(-1) vectors. We additionally normalize the vectors by dividing them by $\sqrt{d}$, where $d$ is the dimension of the vectors. Note that in this case since $\sum_{i=1}^{10} \mathbf{x}_i = \mathbf{0}$ we have $R_2/R_1 = -1$ following (83). We then change the signs of the elements (one at a time) of the 5 all-$(-1/\sqrt{d})$ vectors, until they are all-$(1/\sqrt{d})$. Note that at the final iteration we have $\mathbf{x}_1 = \mathbf{x}_2 = \ldots \mathbf{x}_n$ which corresponds to $R_2/R_1 = n - 1 = 9$. The algorithm for varying $R_2/R_1$ and estimating the MSE at each step is shown below,

---

**Algorithm 1:** Estimating MSE for varying $R_2/R_1$

**Input:** $n$ (even), $k$, dimension $d$, number of iterations $T$, ESTIMATOR (from Rand-$k$-Spatial family)

**Data:** $\mathbf{x}_i = \frac{1}{\sqrt{d}}$ for $i \in [1, \frac{n}{2}]$, $\mathbf{x}_i = \frac{-1}{\sqrt{d}}$ for $i \in [\frac{n}{2} + 1, n]$

1 **for** $j = 1, 2, \ldots, d$ **do**
2    **for** $i = \frac{n}{2} + 1, \ldots, n$ **do**
3      Set $s = 0$
4      Set $x_{ij} = \frac{1}{\sqrt{d}}$
5      **for** $t = 1, 2, \ldots, T$ **do**
6        Sample $\mathbf{h}_1, \mathbf{h}_2, \ldots \mathbf{h}_n$
7        $\hat{\mathbf{x}} = \text{ESTIMATOR}(\mathbf{h}_1, \mathbf{h}_2, \ldots, \mathbf{h}_n)$
8        $s = s + \|\hat{\mathbf{x}} - \bar{\mathbf{x}}\|^2$
9      **end**
     **Output:** Estimated MSE = $\frac{s}{T}$
10    **end**
11 **end**

---

We present here additional results for $k = 1$ and $k = 50$ keeping $n = 10, d = 100, T = 1000$ fixed in Fig. 6.

We note that as $k/d$ increases the relative difference between the performance of various estimators starts increasing. In particular, for $k/d = 0.1$ we see that the MSE of all estimators is of the same order. However, on increasing $k/d$ to 0.5, we see that Rand-$k$-Spatial(Max) and Rand-$k$-Spatial(Avg.) have an order of magnitude lower MSE than Rand-$k$ for $R_2/R_1 \geq 8$. This further strengthens our proposition for using the Rand-$k$-Spatial family of estimators especially at high values of $k/d$.

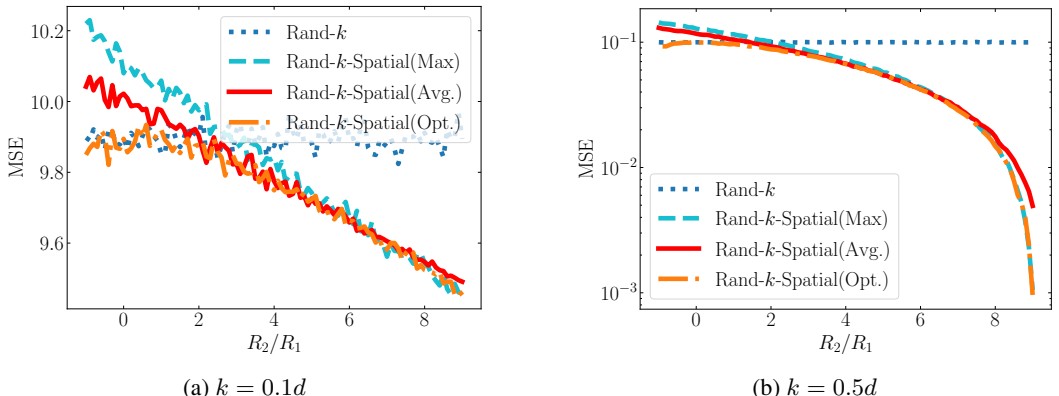

(a) $k = 0.1d$

(b) $k = 0.5d$

Figure 6: MSE v/s $R_2/R_1$ for the Rand-$k$-Spatial family of estimators for $k = 0.1d$ and $k = 0.5d$. Rand-$k$-Spatial(Avg.) closely matches the performance of the optimal estimator across the range of $R_2/R_1$ in both cases. Moreover, as $k/d$ increases the MSE of the Rand-$k$-Spatial family of estimators relative to Rand-$k$ improves by an order of magnitude for high values of $R_2/R_1$.

## B.2 $R_2/R_1$ **for real-world data settings**

We demonstrate how $R_2/R_1$ varies across iteration when nodes are performing some real-world tasks under different data settings. We focus on distributed Power Iteration which uses mean estimation as a subtask. We partition the Fashion-MNIST and CIFAR-10 datasets equally among 100 nodes, either in an IID or Non-IID fashion and plot how $R_2/R_1$ varies for the vectors generated at the nodes in each round. For more details on the dataset and data split please refer to Appendix C.

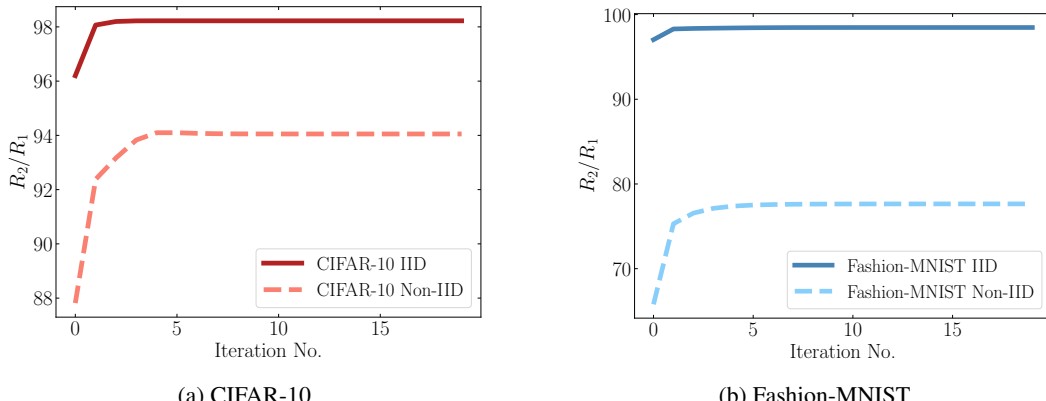

(a) CIFAR-10             (b) Fashion-MNIST

Figure 7: $R_2/R_1$ across iterations for nodes performing distributed Power Iteration under different data settings. In the IID case $R_2/R_1$ quickly reaches the maximum possible value (99 in this case), indicating that node vectors converge to the same point. In the Non-IID case we see a drop in $R_2/R_1$ which is expected as the convergence of node vectors is now dependent on their local data. However $R_2/R_1$ is still significantly greater than 0, pointing to the benefit of using the Rand-$k$-Spatial(Avg.) estimator in such settings. This is demonstrated by our experimental results in Fig. 8.

Our results show that $R_2/R_1$ is likely to vary and be greater than zero in real world settings, thus further motivating the use of Rand-$k$-Spatial(Avg.) estimator. We note that in the IID case, node vectors get highly correlated after a few rounds as $R_2/R_1$ reaches close to the maximum of 99 indicating that $\mathbf{x}_1 \approx \mathbf{x}_2 \cdots \approx \mathbf{x}_n$. In the Non-IID setting, we see a drop in $R_2/R_1$ indicating that node vectors are more dissimilar at convergence, which is expected. However, $R_2/R_1$ is still higher than $n/2 = 50$ in which we case we expect Rand-$k$-Spatial(Avg.) to outperform the Rand-$k$ estimator. This is corroborated by our experimental findings in Fig. 8.

# C   Experimental Details

## C.1   Platform

Experiments were run on Google Colab, a free cloud service providing interactive `Jupyter` notebooks to run and execute `Python` code through the browser. We use `Numpy` for implementing our algorithms.

## C.2   Dataset Description:

The Fashion-MNIST(FMNIST) dataset consists of 60,000 training images (and 10,000 test images) of fashion and clothing items, taken from 10 classes (7000 images per class). The dimension of each image is 28×28 in grayscale (784 total pixels). Fashion-MNIST is intended to be used as a compatible replacement for the original MNIST dataset of handwritten digits.

The CIFAR-10 dataset is a natural image dataset consisting of 60000 32x32 colour images, with each image assigned to one of 10 classes (6000 images per class). The data is split into 50000 training images and 10000 test images.

## C.3   Data Split:

For Non-IID data split we follow a similar procedure as [25]. The data is first sorted by labels and then divided into $2n$ shards with each shard corresponding to data of a particular label. Each of the $n$ nodes is then assigned 2 such shards. For the IID split nodes are assigned data chosen uniformly at random from the dataset. Note that in all cases, data is partitioned equally among all nodes.

## C.4   Experiments

We focus on the following three applications of our proposed sparsification techniques that use mean estimation as a subroutine i) *Power Iteration* ii) *K-Means* iii) *Logistic Regression*. We present below additional details and results for each of the three applications by varying the compression parameter $k$ and incorporating different data settings.

### i) Power Iteration:

The goal of the server here is to estimate the principal eigenvector of the covariance matrix of the data distributed across 100 nodes by power iteration. Given a current estimate of the eigenvector, nodes perform one step of power iteration on their local covariance matrix and send back the updated eigenvectors to the server. These updates are then averaged by the server and normalized to form a new estimate of the eigenvector for the next round. The initial estimate of the principal eigenvector is drawn from $[0,1]^d$. We present here additional results for $k = \{0.05d, 0.1d, 0.2d\}$ both for IID and Non-IID data splits on the Fashion-MNIST dataset. Our results show that Rand-$k$-Spatial(Avg.) and Rand-$k$-Temporal *significantly* outperform other baselines in *all settings*.

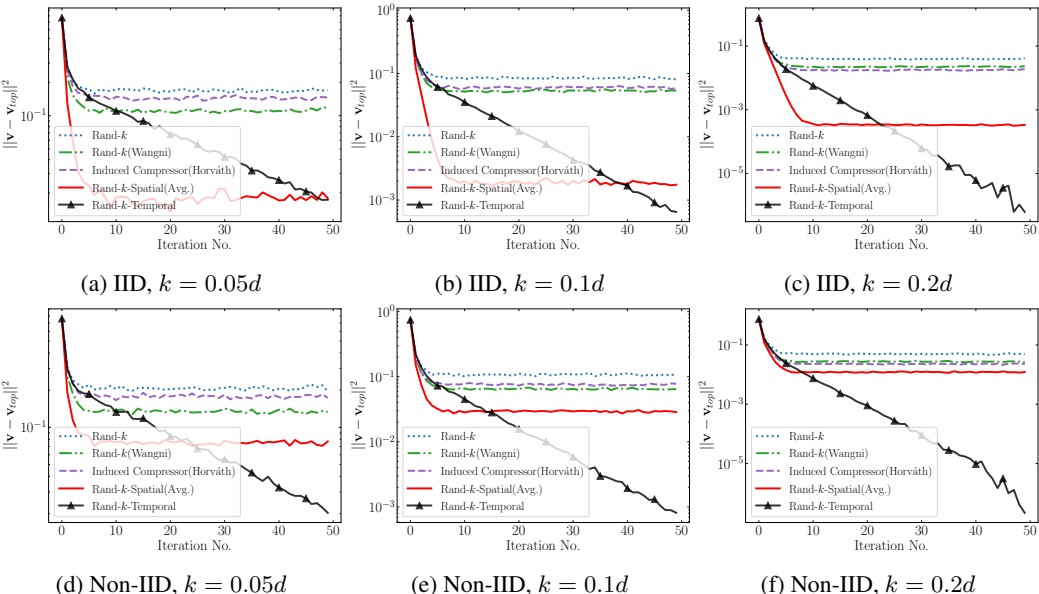

Figure 8: Experiments on distributed Power Iteration on Fashion-MNIST dataset for varying $k$ under different data splits. Note that Rand-$k$-Spatial(Avg.) and Rand-$k$-Temporal outperform baselines in all cases. While Rand-$k$-Temporal performs equally well in IID and Non-IID settings, the performance of Rand-$k$-Spatial is affected by the data split. In particular, Rand-$k$-Spatial(Avg.) performs best in the IID data setting, where we expect spatial correlation to be higher.

## ii) K-Means:

The goal of the server here is to cluster data points distributed across 100 nodes into 10 different clusters using Llyod's algorithm. Nodes update the current cluster centres based on their local data and send back the updated centres to the server. The server then computes a weighted average of the updated centres for each cluster. Note that this effectively reduces to solving 10 different instances of the mean estimation problem. Cluster centers are initialized by randomly assigning them to one of the node data points. We present here additional results for $k = \{0.05d, 0.1d, 0.2d\}$ for both IID and Non-IID data splits on the Fashion-MNIST dataset. We see that Rand-$k$-Temporal outperforms baselines in most cases, while Rand-$k$-Spatial outperforms baselines in the IID case and performs equally well in the Non-IID case where spatial correlation is expected to be lower.

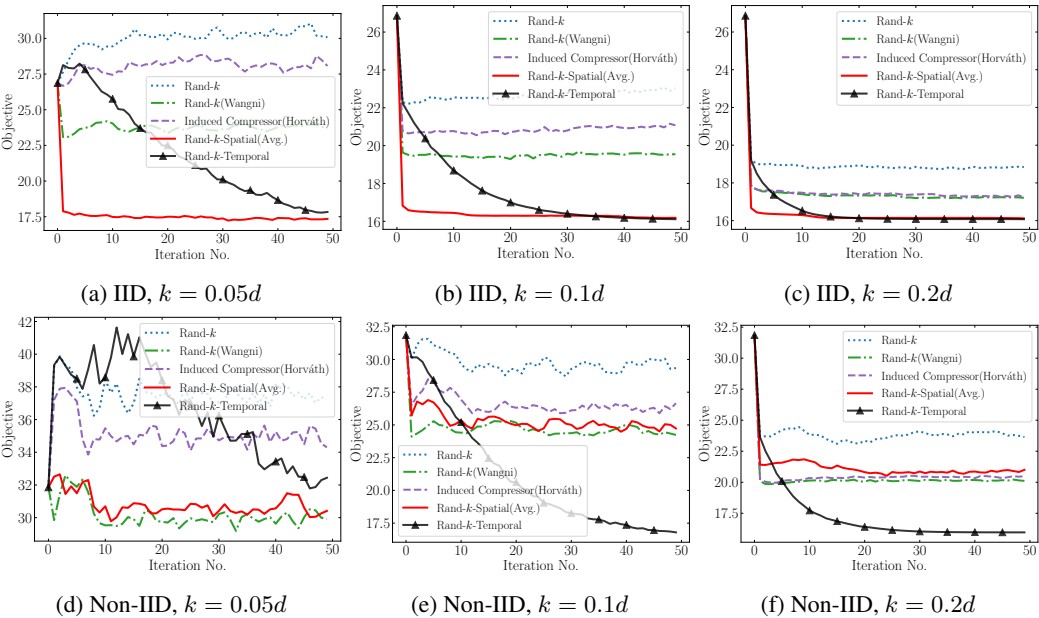

(a) IID, $k = 0.05d$      (b) IID, $k = 0.1d$      (c) IID, $k = 0.2d$

(d) Non-IID, $k = 0.05d$      (e) Non-IID, $k = 0.1d$      (f) Non-IID, $k = 0.2d$

Figure 9: Experiments on distributed K-Means on Fashion-MNIST dataset for varying $k$ under different data splits. Note that we solve 10 instances of the mean estimation problem for each of the cluster centres which makes compression schemes more sensitive to the choice of the compression factor $k/d$. This is seen by the fact that baselines start to diverge at $k = 0.05d$. Note that Rand-$k$-Temporal outperforms other methods in most cases while Rand-$k$-Spatial(Avg.) outperforms baselines in the IID case and performs equally well in the Non-IID case where spatial correlation is expected to be lower.

## ii) Logistic Regression:

The goal of the server here is to classify data points distributed across 10 nodes into 10 different classes using a simple linear classifier followed by softmax activation. Nodes compute stochastic gradients on the global model sent by the central server on their local data, which is then sparsified and aggregated at the central server to update the global model. We use a learning rate $\eta = 0.01$ and batch size of 512 at the nodes. We present here additional results for $k = \{0.005d, 0.01d, 0.02d\}$ for a Non-IID data split on the CIFAR-10 dataset showing both the training loss and test accuracy. We use a much higher compression factor here as we find node vectors are more robust to compression compared to Power Iteration and K-Means.

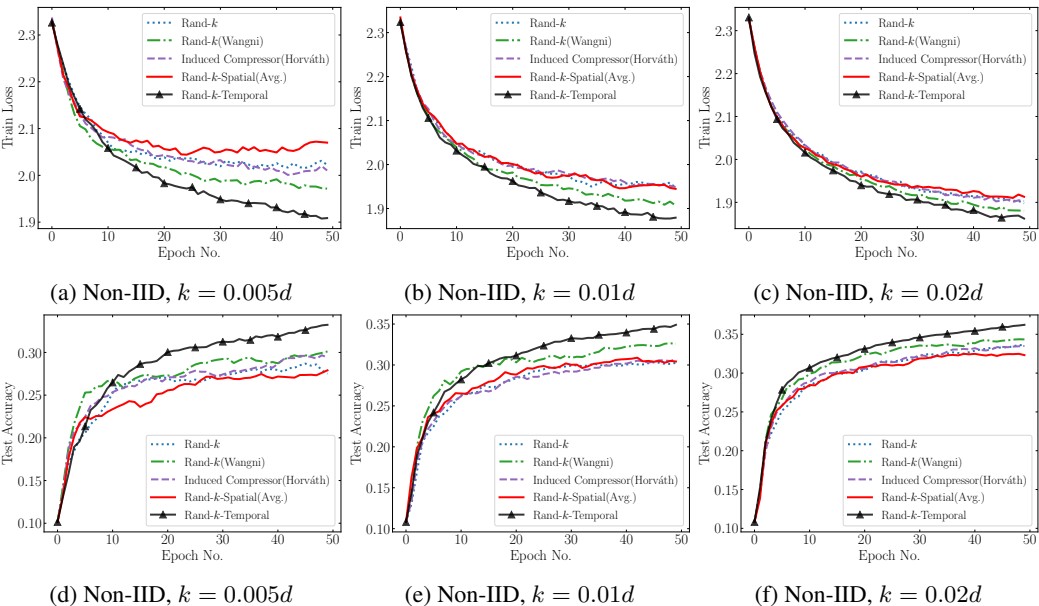

(a) Non-IID, $k = 0.005d$      (b) Non-IID, $k = 0.01d$      (c) Non-IID, $k = 0.02d$

(d) Non-IID, $k = 0.005d$      (e) Non-IID, $k = 0.01d$      (f) Non-IID, $k = 0.02d$

Figure 10: Experiments for logistic regression on CIFAR-10 dataset distributed across 10 nodes. Observe that Rand-$k$-Temporal substantially outperforms other baselines, and achieves lower training loss and 2-3% higher test accuracy in all cases.

We see that Rand-$k$-Temporal substantially outperforms other baselines, and achieves lower training loss and 2-3% higher test accuracy in all cases of compression factors. Interestingly, the performance of Rand-$k$-Spatial seems to be affected due to the non-IID split, which opens up an interesting direction for future work.

# D  Additional Experiments:

## D.1  Reducing Storage Cost at the Server for Rand-$k$-Temporal:

As outlined in our discussion on reducing the storage cost of the Rand-$k$-Temporal estimator, we propose to keep the vector $\mathbf{b}_i$ fixed for all $i \in [n]$, thereby reducing the storage cost to just $\mathcal{O}(d)$. More specifically, we set $\mathbf{b}_i^{(t+1)} = \hat{\mathbf{x}}^{(t)}$ for all $i \in [n]$ where $\hat{\mathbf{x}}^{(t)}$ is the mean estimate at round $t$ (assuming $\mathbf{b}^{(0)} = \mathbf{0}$). Experimental results for this $\mathcal{O}(d)$ memory strategy for distributed Power Iteration experiment on Fashion-MNIST with IID data and $n = 100$ nodes are shown below. Interestingly we observe that this strategy achieves a lower error floor at much fewer iterations compared to all other sparsification and estimation strategies, especially for smaller values of $k$. A drawback however of reducing storage is that the error floor does not decay to zero, as is the case with Rand-$k$-Temporal with full storage. This points to a non-trivial trade-off between storage cost and mean estimation error, which we believe is an interesting direction for future work.

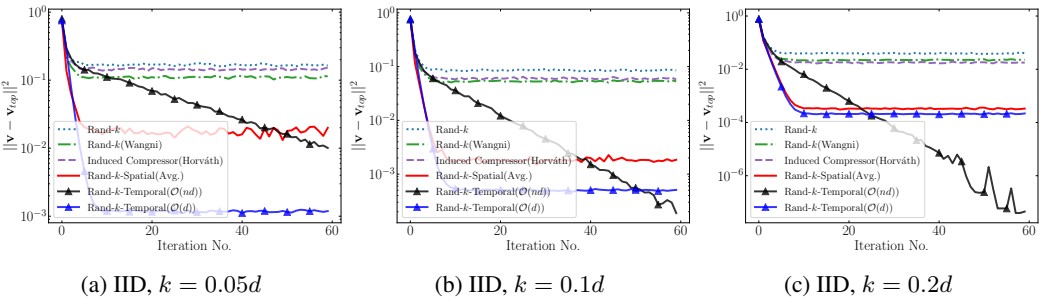

(a) IID, $k = 0.05d$    (b) IID, $k = 0.1d$    (c) IID, $k = 0.2d$

Figure 11: Experiments on distributed Power Iteration on Fashion-MNIST dataset for varying $k$ with IID data split, comparing the performance of Rand-$k$-Temporal with $\mathcal{O}(d)$ memory against other sparsification and estimation strategies. Observe that despite the reduced storage, this strategy continues to achieve a lower error floor than other sparsification strategies that do not utilize temporal information, thereby confirming the effectiveness of our approach.