# OpenReview forum: "Leveraging Spatial and Temporal Correlations in Sparsified Mean Estimation"
_NeurIPS.cc/2021/Conference — NeurIPS 2021 Poster_

### Official Review · Reviewer_87xK · 2021-07-15

**Rating:** 7
**Confidence:** 3

**Summary:**

This paper proposed new estimator that leverages spatial and temporal correlations between vectors in randomized k-sparsified mean estimation procedures.

**Ethics Review Area:**

["I don’t know"]

**Limitations And Societal Impact:**

N.A.

**Main Review:**

This paper proposed new estimator that leverages spatial and temporal correlations between vectors in randomized k-sparsified mean estimation procedures. The presentation is easy to follow, and the ideas are clearly described. The intuition behind the new estimator that utilizes the spatial and temporal correlations is sound under the distriubted learning senario, and the proposed estimator is very simple to understand and implement. Theoretical analysis are presented which clearly show under the strong spatial and temporal correlation cases, the new estimator significantly improves the standard randomized k-sparsified estimator.  Empiricla studies on various datasets and models demonstrated that the proposed technique is a simple and effective way to improve the communication efficiency in distributed learning.

**Time Spent Reviewing:**

N.A.

---

> ### Author Response · Authors · 2021-08-10
> **Response to Reviewer 87xK**
>
> Thank you for the positive comments! We propose two new sparsification strategies- Rand-$k$-Spatial and Rand-$k$-Temporal, utilizing spatial and temporal correlations effectively. As the reviewer points out, our schemes are simple to implement and outperform simple Rand-$k$ as well as other more complex and difficult to implement sparsification strategies in our experiments. Our experimental results are backed by solid theoretical analysis, which clearly show how we incorporate and benefit from spatial and temporal correlation. We look forward to any further questions on our paper!

---

### Official Review · Reviewer_vqXw · 2021-07-16

**Rating:** 6
**Confidence:** 3

**Summary:**

This paper studies the distributed mean estimation problem. In this problem, each node holds a d-dimensional vector and the goal is to compute the mean of all vectors.
The communication cost is the resource to be optimized. The authors propose to exploit spatial and temporal correlations in real world data sets to improve the MSE.
To do so, a new estimate is proposed and analyzed.

**Limitations And Societal Impact:**

Yes.

**Main Review:**


Weakness:
1. My main concern is about the estimate (4). The authors claim without proof that it is unbiased. However, it seems that the authors implicitly assume that $M_j$
is independent of $h_{ij}$. But $M_j$ has non-trivial correlation with $h_{ij}$'s. The same problem happens in the proof of Theorem 1. In the proof, the authors use the law of
total expectation, but for different values of $M_j$, the conditional expectation of $h_{ij}$ are all claimed to be the same, which seems incorrect to me. So the authors should provide
an explanation on this issue in the rebuttal.

2. The second concern is that the estimate (4) is only for the case when all coordinates are sampled with the same probability. However, the uniform sampling scheme often has much worse MSE than schemes using sampling probabilities that depend on the value of each coordinate, e.g., [1].

3. The motivation to utilize spatial and temporal correlations is valid, however the solution proposed is quite simple (if not straightforward).
So I think the technical depth of the paper is below the bar of NeurIPS.

References
[1] D. Alistarh, D. Grubic, J. Li, R. Tomioka, and M. Vojnovic. Qsgd: Communication-efficient sgd via gradient quantization and encoding

**Time Spent Reviewing:**

4

---

> ### Author Response · Authors · 2021-08-10
> **Response to Reviewer vqXw**
>
> Thank you for the review. We address your specific concerns below. Please consider increasing your score if your concerns have been answered below!
>
> **1. Concern about unbiasedness of estimate (4):**
>
> The estimate (4) is indeed unbiased, as the paper claims. The misunderstanding might have been caused due to a slight typo in our definition of the scaling factor $\bar{\beta}$ and we apologize for the confusion.
>
> Recalling the notation from our paper, $\mathbf{h_i}$ is a sparsified version of the device vector $\mathbf{x_i}$ and $h_{ij}$ represents the $j^{th}$ coordinate of $\mathbf{h_i}$. We denote the dimension of $\mathbf{x_i}$ by $d$ and the number of sampled coordinates by $k$. $M_j$ is a random variable denoting the number of nodes that send their $j^{th}$ coordinate. We define,
>
>  $\bar{\beta}= ( \frac{k}{d} \mathbb{E}_{M_j|M_j \geq 1} [\frac{1}{T(M_j)}] )^{-1}.$
>
> Note the expectation is with respect to $M_j|M_j \geq 1$ rather than $M_j$. We do not assume that $M_j$ is independent of $h_{ij}$, as we show in our proof below (we will add this proof in the supplementary material).
>
> Let $\xi_{ij}$ be an indicator random variable, which is $1$ or $0$ depending on whether $h_{ij} = x_{ij}$ or not.
>
>
> * Case 1: With probability $(1-\frac{k}{d})$, $\xi_{ij} = 0$ which implies $h_{ij} = 0$. Therefore,
>
> $\mathbb{E}_{M_j|\xi\_{ij}=0}\left[\frac{\bar{\beta}h\_{ij}}{T(M_j)}\right] = 0 $
>
>
> * Case 2:  With probability $\frac{k}{d}$, $\xi_{ij} = 1$ which implies $h_{ij} = x_{ij}$. Therefore,
>
> $\mathbb{E}_{M_j|\xi\_{ij}=1}\left[\frac{\bar{\beta}h\_{ij}}{T(M_j)}\right]  = \mathbb{E}\_{M_j|M_j \geq 1} \left[\frac{\bar{\beta}h\_{ij}}{T(M_j)}\right] = \bar{\beta}x\_{ij}\mathbb{E}\_{M_j|M_j \geq 1} \left[\frac{1}{T(M_j)}\right]$
>
> The crucial observation here is that $\xi_{ij} = 1$ only implies $M_j \geq 1$ and does not give any other information about $M_j$. This allows us to decouple the relation between $h_{ij}$ and $M_j$ for our unbiasedness proof as well as our proof of Theorem 1. Next, taking expectation with respect to $\xi_{ij}$ we have,
>
> $\mathbb{E}_{\xi\_{ij}} \mathbb{E}\_{M_j|\xi\_{ij}} = \frac{k}{d} \bar{\beta}x\_{ij}\mathbb{E}\_{M_j|M_j \geq 1} \left[\frac{1}{T(M_j)}\right] = x\_{ij},$
>
> which follows from the definition of $\bar{\beta}$. This proves that that estimate (4) is unbiased.
>
> **2. Uniform sampling versus Non-uniform sampling depending on the value of each coordinate:**
>
> Firstly, we note that the reference [1] mentioned by the reviewer is using a quantization strategy while the focus of our paper is on sparsification strategies. Quantization is an orthogonal direction to sparsification and can be applied on top of sparsification for additional communication savings.
>
> In case the reviewer meant to suggest non-uniform sampling in the context of sparsification, we would like to highlight the comparison with Rand-$k$(Wangni) [2], a sparsification strategy that samples each coordinate according to its magnitude, that is presented in our paper. Our experiments show that Rand-$k$-Spatial and Rand-$k$-Temporal clearly outperform Rand-$k$(Wangni) even while using uniform coordinate sampling, especially for Power Iteration (Fig:5(a)) and K-Means (Fig:5(b)). Moreover, coordinate dependent sampling strategies such as Rand-$k$(Wangni) incur an additional computation cost at the devices, which our proposed schemes avoid.
>
> **3. Simplicity and technical depth of our paper:**
>
> We respectfully disagree with the reviewer here. As pointed out by the other reviewers, the novelty and simplicity of our proposed solutions is a major contribution, especially when we consider participating devices to be resource-constrained. It is not immediately clear how we can benefit from spatial and temporal correlations in such a setting and we believe we are the first ones to show how such correlation-aware estimation strategies can be designed. Moreover the simplicity of implementation of our proposed estimators, should not be used to measure technical depth as our analysis is non-trivial and provides deep theoretical insights into the performance of the proposed solutions as well as the design of other correlation-aware strategies.
>
> **References**
>
> [1] Alistarh, Dan, et al. "QSGD: Communication-efficient SGD via gradient quantization and encoding." _Advances in Neural Information Processing Systems_ 30 (2017): 1709-1720.
>
> [2] Wangni, Jianqiao, et al. "Gradient sparsification for communication-efficient distributed optimization." _arXiv preprint arXiv:1710.09854_ (2017).

---

> > ### Comment · Reviewer_vqXw · 2021-08-28
> > **Response to rebuttal**
> >
> > Dear authors,
> >
> > Thanks for the detailed response.  Now I am convinced that the error raised in my original review is fixable and I increase my evaluation. Overall I think the paper contains some novel insights although the techniques used are simple. I still think uniform sampling is somewhat restrictive. For example, for skewed vectors non-uniform sampling could be much better. It seems that how to extend the proposed estimators to handle non-uniform sampling is not easy.

---

> > > ### Author Response · Authors · 2021-09-02
> > > **Thank you very much for the increased evaluation!**
> > >
> > > Thank you for your response to our rebuttal. We are very grateful for the increased rating!
> > >
> > > We chose to analyze our proposed estimator with Rand-$k$(uniform sampling) due to its simplicity and ease of exposition. Our proposed estimators can also be applied with non-uniform sampling techniques such as Top-$k$ and Rand-$k$(Wangni), albeit at the cost of additional computation at nodes and possible bias in estimation. As you mentioned, this can potentially give better performance for skewed vectors and we believe analyzing such setups is an interesting direction for future work.

---

### Official Review · Reviewer_6xmC · 2021-07-16

**Rating:** 5
**Confidence:** 3

**Summary:**

This paper studies the problem of distributed "Mean Estimation". Authors provide new insight for the sparsified approach and propose a new method that exploits the data correlation. They show that theoretically, their proposed approach for the special cases of random K outperforms the available technique. Finally, theoretical findings are supported by experiments.

**Limitations And Societal Impact:**

See above comments!

**Main Review:**

Positive aspects:

- I think the paper is well-written and it is easy to follow.
- There is some novelty in the way authors try to use the correlation in the data to increase the sparsity of transmitted vectors.
- Experiments look reasonable.

Downsides:

- While Theorem 2 provides some theoretical insight regarding the connection between data correlation and optimal sparsification scheme, I think the result in the distributed setting is not useful as it depends on the cross product of the date which is not available at the server. So, It is difficult to consider this result as remarkable.

- My major concern about this paper is the additional storage cost required at the server. In essence, the server is iteratively is trying to reconstruct the data at the devices, which has three major issues when it comes to applications such as Federated Learning as pointed out by authors. First, in FL setting there could be a huge number of devices, therefore storing a vector associated with each device could be impractical and also this brings us to the second point that if the server can store as many as vectors why do we need to do this process distributedly. In other words, given this extra storage at the server, devices can send their data and the server can easily take the average. Third (and least importantly), roughly speaking as vectors $b_i$ represents the data point at the server, this could be considered as the violation of data privacy by the server, especially in Federated Learning.

**Time Spent Reviewing:**

4 hours more or less

---

> ### Author Response · Authors · 2021-08-10
> **Response to Reviewer 6xmC**
>
> Thank you for appreciating the writing in our paper and for highlighting the novelty of our proposed schemes. We address all the specific concerns below. Please consider increasing your score if your questions have been answered below!
>
> **1. Connection between data correlation and optimal sparsification scheme:**
>
> We would like to stress that the exact value of the cross-product is only needed to design the *optimal* Rand-$k$-Spatial estimator. In practice, since the data correlation is unknown, the estimated cross-product ratio $R_2/R_1$ can be treated as a hyperparameter and tuned using standard search strategies. Furthermore, to avoid the cost of tuning this hyperparameter altogether, we propose the Rand-$k$-Spatial(Avg.) estimator (see page 5), which sets the estimated cross-product ratio $R_2/R_1$ to $n/2$. Our experimental results in Fig:5(a)-(f) show that Rand-$k$-Spatial(Avg.) is a good default choice and offers a clear improvement over Rand-$k$ in most cases.
>
> We believe that the more important theoretical contribution is our result in Theorem 1, which shows that the Rand-$k$-Spatial estimator can effectively utilize correlation among device vectors to reduce the mean estimation error. This opens up the possibility of designing the optimal spatial estimator utilizing knowledge of device vector correlation, as we show in Theorem 2.
>
>
> **2. Concerns about Storage Cost at the server:**
>
> In the naive implementation of the proposed Rand-$k$-Temporal estimator, we need $O(n d)$ storage/memory at the server to store previously sent vector $\mathbf{b}_i$ from each device $i$. However, this memory cost can be easily reduced. Our result in Theorem 3 holds for arbitrary $\mathbf{b}_i$ and thus the $\mathbf{b}_i$ do not have to be necessarily different for each device. Instead, we can set them using one of the following strategies:
>
> * A natural idea to reduce the memory requirement at the server to just $O(d)$ is to set $\mathbf{b}_i = \mathbf{x}^{(t)}$ for all $i \in [n]$, where $\mathbf{x}^{(t)}$ is the previous mean estimate. Thus, we store a single reference vector for all devices. We tested this $O(d)$ memory strategy for Distributed Power Iteration experiment on Fashion-MNIST with IID data, extending the results in Fig: 5(a), as we show below. Observe that even with $O(d)$ memory, we get similar performance to the naive $O(n d)$ memory implementation Rand-$k$-Temporal estimator.
>
> | Sparsification Method     | Distance to principal eigenvector(after 60 rounds)  |
> | :---        |    :----:   |
> | Rand-$k$     | 0.082617  |
> | Induced Compressor[2]   | 0.058570 |
> | Rand-$k$(Wangni)[3]  | 0.051951  |
> | Rand-$k$-Spatial(Avg.)    | 0.002040 |
> | Rand-$k$-Temporal($\mathbf{b}_i = \mathbf{x}^{(t)}$ )  | 0.000512  |
> | Rand-$k$-Temporal  | 0.000317  |
>
> *  An alternative is to perform $c$-means clustering on previously sent data vectors and only store the $c$ cluster centers at the server, thus reducing the memory cost from $O(nd)$ to $O(c d)$.
>
> * Finally, instead of keeping a memory at the server, each device can also locally store $\mathbf{b}_i$, thus distributing the $O(n \times d)$ storage cost at the server to an $O(d)$ storage cost at each of the $n$ devices. We chose server-side memory as it avoids burdening resource-constrained devices, and instead puts the burden on the central server which is assumed to have significantly higher storage and computational capacity.
>
> **3. Concerns about Data privacy:**
>
>  We believe that the reviewer may have a misunderstanding about the privacy properties of the Rand-$k$-Temporal estimator. The confusion may stem from our use of the term "data vectors" for the $\mathbf{b}_i$ vectors sent by the devices to the central server. In the context of federated learning, these *vectors would be gradients or model updates computed by each device and NOT the training data samples used to obtain the gradients*. The Rand-$k$-Temporal estimator offers the same privacy guarantees in Federated Learning as any other standard sparsification and estimation strategy. The only difference here is that we choose to store and reuse previously communicated gradients or model updates. We believe this does not constitute any major privacy concern. To be clear, we are *not* trying to reconstruct local device data at the server. Please let us know if you have any further questions!

---

> > ### Author Response · Authors · 2021-09-02
> > **We would appreciate any feedback to our response.**
> >
> > Dear Reviewer 6xmC,
> >
> > Thank you again for your review.
> >
> > We believe we have addressed the concerns you raised about our work, especially those related to storage cost and privacy of the Rand-$k$-Temporal estimator. If so, we would be very grateful if you could kindly consider increasing your rating of our work. We are quite happy to discuss further if you have more questions. Thank you!

---

### Official Review · Reviewer_3MWC · 2021-07-16

**Rating:** 6
**Confidence:** 4

**Summary:**

The paper proposes two new sparsification methods for the distributed mean estimation problem. One of the proposed strategies, Rand-k-Spatial, uses the spatial correlation of different nodes to have an unbiased estimator. The other strategy, Rand-k-Temporal, uses the temporal correlation of local vectors at consecutive iterations to fill the sparsified elements without an extra computation or storage cost at the resource-constrained nodes. The authors show that both methods outperform Random-k and Top-k sparsification methods empirically.

**Limitations And Societal Impact:**

The authors do not mention limitations or the potential negative societal impact of their work.

**Main Review:**

Overall, I liked the methodology and enjoyed reading the paper. Although the proposed methods might seem incremental upon Random-k, the way authors get use of temporal and spatial correlations to improve Random-k is neat and theoretically sound. I also agree with the authors that the approach is applicable to other sparsification methods as well. Here, I list some more specific comments and questions:

- The paper refers to top-k and random-k frequently and compares the methods' strengths and weaknesses. However, I think they miss a strategy, rTop-k [1], that combines these two methods and enjoys both top-k's and random-k's benefits.

- Although the concept of spatial correlations is not new, temporal correlations have recently started getting attraction. And I like the way how authors benefit from temporal correlations.

- It is interesting that the authors were able to outperform top-k with the modified version of random-k. The discussion on rand-k-temporal and error feedback seems important to me given that many sparsification methods do not perform well without error feedback, hence they suffer from high storage costs at the nodes.

- Do the authors have an explanation as to why rand-k-spatial does not perform well in logistic regression experiments in Figures 5(c,f)? Is this because the data is non-IID for that experiment? Then, does that mean that rand-k-spatial does not work well with non-IID data? If so, then the proposed method is not very useful for practical federated learning settings. Can the authors clarify if I am missing something here?

[1] Barnes, Leighton Pate, et al. "rTop-k: A statistical estimation approach to distributed SGD." IEEE Journal on Selected Areas in Information Theory 1.3 (2020): 897-907.

**Time Spent Reviewing:**

3 hours

---

> ### Author Response · Authors · 2021-08-10
> **Response to Reviewer 3MWC**
>
> Thank you for the positive review and for highlighting the methodology and theoretical soundness of our proposed schemes. We address each of your specific comments below. We would request the reviewer to kindly raise their score if all concerns have been met satisfactorily.
>
> **1. Comparison to rTop-k:**
>
> Thank you for pointing us to rTop-k which is indeed a valuable and apt reference strategy for our proposed schemes. We will add a reference to rTop-k in the final version of our paper. To compare rTop-k with our schemes, we ran experiments for Distributed Power Iteration on Fashion-MNIST with IID data, extending the results in Fig. 5(a), as we show below. We use the same $k$ for rTop-k as other sparsification strategies and set $r=2k$ (tuned using a grid search) along with the use of error feedback.
>
>
> | Sparsification Method     | Distance to principal eigenvector(after 60 rounds)  |
> | :---        |    :----:   |
> | Rand-$k$     | 0.082617  |
> | Induced Compressor[2]   | 0.058570 |
> | rTop-k   | 0.057964  |
> | Rand-$k$(Wangni)[3]  | 0.051951  |
> | Rand-$k$-Spatial(Avg.)    | 0.002040 |
> | Rand-$k$-Temporal  | 0.000317  |
>
> While rTop-k does significantly better than Rand-$k$ and is comparable to the Induced Compressor, it is outperformed by the proposed Rand-$k$-Spatial(Avg.) and Rand-$k$-Temporal estimators due to their use of spatial and temporal correlations respectively, something which rTop-k does not account for. As the reviewer pointed out, our approach can also be applied on top of other sparsification strategies including rTop-k and we believe doing so will improve general performance.
>
>
> **2. Novel use of temporal correlation:**
>
> Thank you for highlighting this contribution! The theoretical analysis of spatial correlations for distributed mean estimation is also novel to the best of our knowledge.
>
>
>
> **3. Discussion on error feedback and storage cost:**
>
> Thank you for highlighting this point! Indeed, biased sparsification strategies do not do well without error feedback and require an extra storage cost at the nodes, which Rand-$k$-Temporal estimator avoids while being unbiased.
>
>
> **4. Rand-$k$-Spatial does not perform well in Logistic Regression for non-IID data:**
>
> Rand-$k$-Spatial(Avg.) indeed does not do well in the Logistic Regression experiments due to the non-IID split of data. However, for experiments on Power Iteration and K-Means with non-IID split, Rand-$k$-Spatial(Avg.) continues to outperform other sparsification strategies in most cases as shown by experimental results Fig 8:(d)-(f) and Fig 9:(d)-(f) in our supplementary material. The reason for this is that the performance of the Rand-$k$-Spatial estimator depends on the device vector cross-product ratio $R_2/R_1$, which Rand-$k$-Spatial(Avg.) assumes to be $n/2$ as a default choice.
>
> On calculating the actual $R_2/R_1$ value averaged over first 10 rounds ($n= 100$ devices) with non-IID data split, we find the value to be $23.42$ for Logistic Regression and $75.71$ for Power Iteration, implying that device vectors are much less correlated in the logistic regression setting. We can use a lower estimate of $R_2/R_1$ as a default choice for gradient based settings for better performance. Setting $R_2/R_1 = 0$ in the worst case reduces to the Rand-$k$ estimator. Interestingly, we observe that the value of $R_2/R_1$ is significantly increased in logistic regression when devices send model weights instead of model gradients with $R_2/R_1 = 98.19$. We believe this is of importance for optimization algorithms such as *Stochastic Gradient Push*[4] and *Decentralized Parallel SGD* [5] which perform averaging of model weights rather than model gradients.
>
> **References:**
>
> [1] Barnes, Leighton Pate, et al. "rTop-k: A statistical estimation approach to distributed SGD." IEEE Journal on Selected Areas in Information Theory 1.3 (2020): 897-907.
>
> [2] Horváth, Samuel, and Peter Richtárik. "A Better Alternative to Error Feedback for Communication-Efficient Distributed Learning." arXiv preprint arXiv:2006.11077 (2020).
>
> [3] Wangni, Jianqiao, et al. "Gradient sparsification for communication-efficient distributed optimization." arXiv preprint arXiv:1710.09854 (2017).
>
> [4] Nedić, Angelia, and Alex Olshevsky. "Stochastic gradient-push for strongly convex functions on time-varying directed graphs." IEEE Transactions on Automatic Control 61.12 (2016): 3936-3947.
>
> [5] Lian, Xiangru, et al. "Can decentralized algorithms outperform centralized algorithms? a case study for decentralized parallel stochastic gradient descent." arXiv preprint arXiv:1705.09056 (2017).

---

> > ### Comment · Reviewer_3MWC · 2021-08-27
> > **response to rebuttal**
> >
> > I would like to thank the authors for the additional experimental results in the rebuttal. Overall, I think the paper has promising findings and I keep my original rating.

---

### Author Response · Authors · 2021-08-27
**We would appreciate any feedback to our response and opportunity for further clarification.**

We thank Reviewer 3MWC for responding to our rebuttal and for highlighting the relevance of the findings discussed in our paper.

We would greatly appreciate it if other reviewers can let us know whether our response has addressed their concerns, especially those related to storage and privacy (as pointed out by Reviewer 6xmC ) and unbiasedness of estimate (4) (as pointed out by Reviewer vqXw). We are quite willing to clarify further if more questions arise. Thank you!

---

### Decision · Program_Chairs · 2021-09-27

**Decision:**

Accept (Poster)

**Comment:**

This paper studies communication efficient estimators for the distributed mean estimation problem. The paper presents two methods that exploit either spatial or temporal correlations of the data, with the goal to improve communication efficiency. The MSE of the two estimators is studied analytically, and numerical benchmarks demonstrate that the proposed techniques can improve the communication efficiency in distributed learning.

The reviewers spotted a few inaccuracies in the proofs, but following the author's response they believe that these issues can be addressed in the final version.

On the one hand, the reviewers emphasized the simplicity of the method, but on the other hand, they found the contribution to be
slightly incremental. At the end, the good experimental results were the deciding factor in the discussion.

The reviewers believe that this work inspire future work (including attempts to address current limitations regarding practicality in distributed settings) and will be of interest to the community.

The authors are strongly encouraged to take the reviewer's feedback into account when preparing the final version, including the already proposed changes, and perhaps also to include a discussion on lower bounds from an information-theoretic perspective (as guides for follow-up work).